# Application of Acoustic Cardiography in Assessment of Cardiac Function in Horses with Atrial Fibrillation Before and After Cardioversion

**DOI:** 10.3390/ani15131993

**Published:** 2025-07-07

**Authors:** Mélodie J. Schneider, Isabelle L. Piotrowski, Hannah K. Junge, Glenn van Steenkiste, Ingrid Vernemmen, Gunther van Loon, Colin C. Schwarzwald

**Affiliations:** 1Clinic for Equine Internal Medicine, Vetsuisse Faculty, University of Zurich, 8057 Zurich, Switzerland; ipiotrowski@vetclinics.uzh.ch (I.L.P.); hjunge@vetclinics.uzh.ch (H.K.J.); cschwarzwald@vetclinics.uzh.ch (C.C.S.); 2Equine Cardioteam Ghent, Department of Internal Medicine, Reproduction and Population Medicine, Faculty of Veterinary Medicine, Ghent University, Salisburylaan 133, 9820 Merelbeke, Belgium; glenn.vansteenkiste@ugent.be (G.v.S.); gunther.vanloon@ugent.be (G.v.L.)

**Keywords:** arrhythmia, phonocardiogram, echocardiography, cardiovascular, Audicor^®^

## Abstract

Atrial fibrillation (AF) in horses reduces exercise capacity due to atrial contractile dysfunction, impaired ventricular systolic function, disproportionate ventricular rate response and reduced cardiac output. Atrial dysfunction can persist even after treatment and is considered a risk factor for recurrence. Ambulatory acoustic cardiography (Audicor^®^ (Beaverton, OR, USA)) provides qualitative and quantitative information on hemodynamics and cardiac mechanical function and may aid in the clinical diagnosis of impaired heart function after AF treatment. This study aimed to evaluate the use of an acoustic cardiography monitor to assess cardiac mechanical function and its benefits as an adjunct to echocardiography in horses with AF before and after successful cardioversion to sinus rhythm. Audicor^®^ snapshot analyses provided additional information on left ventricular systolic function and reflected changes seen in echocardiography. However, the Audicor^®^ device did not seem clinically useful as a tool to directly assess left atrial mechanical function after cardioversion to sinus rhythm.

## 1. Introduction

Atrial fibrillation (AF) in horses reduces exercise capacity due to atrial contractile dysfunction, decreased ventricular preload and systolic function, disproportionate ventricular rate response, and irregular heart rhythm, resulting in decreased cardiac output at maximum exercise [1,2,3,4]. Echocardiography at rest reveals atrial and ventricular dysfunction, which generally improves 12 to 24 h after conversion of AF to sinus rhythm [3,5,6,7,8]. However, depending on the duration of AF, left atrial (LA) mechanical dysfunction, also termed “LA stunning”, may persist up to seven weeks after conversion, corresponding to AF-induced atrial remodeling [5,9,10,11,12,13,14]. LA stunning is one of several factors associated with the high AF recurrence (rAF) rate of 15–50% after successful cardioversion [15,16,17,18,19].

In a routine clinical setting, echocardiography remains the only established method for assessing ventricular and atrial size and function in horses, including two-dimensional (2DE) and motion-mode (M-mode) echocardiography, tissue Doppler imaging (TDI), and 2D speckle tracking (2DST) imaging [6,9,10,11,17,20,21,22,23,24,25,26,27,28]. Ambulatory acoustic cardiography (Audicor^®^) is a diagnostic adjunctive to echocardiography in human cardiac patients, offering the advantages of being low in cost, operator-independent, time-efficient, and easy to interpret [29,30]. Several studies in human medicine have confirmed its correlation with echocardiographic indices of cardiac function [29,30,31,32]. Zuber et al. [29] even showed that Audicor^®^ recordings, in particular electro-mechanical activation time (EMAT), were more reproducible in pacemaker-implanted human patients than echocardiographic measurements. Furthermore, EMAT was also an early detector of electromechanical dysfunction before changes in left ventricular ejection fraction (LVEF) became apparent in echocardiography [33]. The Audicor^®^ recording consists of a single-lead surface ECG coupled with a phonocardiogram, allowing quantification of diastolic heart sounds and systolic time intervals [34]. In people, its application ranges from detecting left ventricular (LV) dysfunction [32,34,35,36] and predicting rAF [37,38] to monitoring therapeutic effects [38]. An accentuated third heart sound (S3), an indicator for diastolic function, is considered a risk factor for rAF in humans after successful cardioversion [37]. In contrast to people, both diastolic heart sounds are considered physiological in horses, with S3 being linked to the end of rapid ventricular filling and the fourth heart sound (S4) being linked to atrial contraction [39]. However, heart disease in horses can also cause accentuation or attenuation of diastolic heart sounds, rendering their quantification potentially useful. The feasibility and repeatability of Audicor^®^ analyses in healthy horses as a non-invasive, in-field diagnostic tool to measure cardiac mechanical and hemodynamic function have been previously established [40]. However, the benefits of Audicor^®^ in a clinical setting in horses with heart disease are unclear to date.

This study aimed to examine the use of Audicor^®^ to quantify cardiac mechanical and hemodynamic function in horses with AF before and after successful transvenous electrical cardioversion (TVEC). The objective was to evaluate its use in a clinical setting and its possible benefits as an adjunctive to echocardiography. Specifically, the study investigated whether Audicor^®^ can detect and quantify atrial mechanical dysfunction, impaired ventricular function, as well as recovery thereof after successful restoration of normal sinus rhythm (NSR). To answer these questions, Audicor^®^ recordings and concomitant echocardiographic recordings were compared to evaluate LA and LV function. It was hypothesized that S4 would not be detectable by Audicor^®^ during AF and that the power of S4 would progressively increase upon subsequent follow-up examinations after restoration of NSR, indicating recovery of LA mechanical function. Furthermore, it was postulated that the power of S4 would correlate with echocardiographic variables of LA mechanical function. Lastly, it was hypothesized that Audicor^®^ variables of LV function would improve after cardioversion, mirroring echocardiographic variables of LV function.

## 2. Materials and Methods

### 2.1. Study Design

The study was designed as a multicenter, prospective descriptive study.

### 2.2. Study Sample

Thirty-three horses suffering from AF and successfully treated with TVEC at the Vetsuisse Faculty Zurich (*n* = 4) and at the Faculty of Veterinary Medicine Ghent University (*n* = 29) were recruited. Five horses were excluded due to moderate valvular insufficiencies to exclude hemodynamic consequences of valvular regurgitation as confounding factors. The final study sample consisted of 28 horses (24 Warmbloods, 2 Thoroughbreds, 2 Standardbreds), 6 mares, and 22 geldings, ranging from 3 to 16 years of age (10 ± 4 y [mean ± SD] and a body weight of 584 ± 70 kg. The horses showed trace to mild valvular insufficiencies (5 trace, 11 trivial, and 4 mild aortic insufficiencies; 12 trivial and 11 mild tricuspid insufficiencies; 4 trace, 11 trivial, and 8 mild mitral insufficiencies; 3 trace, 7 trivial, and 2 mild pulmonary insufficiencies) with normal atrial and ventricular dimensions. The use of horses spanned from pleasure to sports horses at different levels of showjumping, dressage, eventing, and racing. All horses stayed in NSR for at least 24 h after conversion. For the 2–7-day follow-up period, only data from horses that remained in NSR throughout this time period were included in the analysis, whereas data from horses that reverted to AF before this time point were excluded.

### 2.3. Study Protocol

All horses underwent a physical examination (demeanor and attitude, body condition score, skin and hair coat, jugular veins, peripheral pulse rate and quality, cardiac auscultation (heart rate (HR), rhythm, murmurs), respiratory rate and lung sounds, mucous membrane color and capillary refill time, gastrointestinal sounds, and rectal temperature), echocardiographic examination, and Audicor^®^ examination at three time points: (1) AF day −1, one day before cardioversion; (2) NSR day 1, one day after cardioversion; and (3) NSR day ≥ 2, two to seven days after cardioversion. Due to the multicenter nature of the study, the third examination was performed at different time points: At the Vetsuisse Faculty Zurich, it was conducted 2 days after cardioversion, while at the Faculty of Veterinary Medicine Ghent University, the time point varied between 3 and 7 days after cardioversion.

### 2.4. Echocardiography

All studies were conducted in standing, unsedated, restrained horses by multiple experienced operators in Zurich and Ghent. The echocardiograms were performed before, during, or after each Audicor^®^ overnight recording (time between beginning of echocardiogram and beginning of Audicor^®^ recording was −26 to +28 h, median 4 h, quartiles 2 to 7 h). At both centers, transthoracic echocardiography was performed with a GE Vivid E95 echocardiograph with a 4Vc-D probe operated in 2D mode and set at a frequency of 1.4/2.8 MHz (GE Healthcare, Freiburg, Germany).

A single-lead base-apex ECG was recorded simultaneously for timing purposes. Recordings were stored as still images or cine-loops in digital raw format for offline analysis (EchoPac v204, GE Healthcare, Freiburg, Germany). Offline measurements were performed by a single operator (MS) after extensive training. Grading of valvular regurgitation, where present, was confirmed by an experienced echocardiographer (CS). Where possible, three representative non-consecutive or consecutive cycles were recorded, measured, and subsequently averaged for each variable. However, in some instances, fewer than three analyzable cardiac cycles were available due to technical limitations during image acquisition, such as suboptimal image quality. The number of measurements based on fewer than three cardiac cycles is summarized in Appendix A. Cycles immediately after very long or very short pauses during AF, after sinus pauses, or after second-degree atrioventricular (AV) blocks were excluded from the analyses. Instantaneous HR was calculated based on the RR interval (ms) preceding the respective measurement:HR = 60,000/RR.

Routine transthoracic 2DE, M-mode, pulse wave TDI (PW TDI) and color TDI (cTDI) echocardiography were performed to assess cardiac structures, valvular competence, chamber dimensions, great vessel dimensions, and LV systolic and diastolic function, by use of standard right and left parasternal long-axis and right parasternal short-axis views [9,20,21,22,25,41,42,43]. The main focus was put on the assessment of LA size and mechanical function using the methods previously described [9,20,22]. Echocardiographic variables and indices used in this study are listed in detail in Appendix A. They included the following: maximum left atrial diameters (LAD_max_ and LAD_llx-max_), maximum left atrial areas (LAA_max_ and LAsxA_max_), left atrial active fractional area change (active LA FAC), left atrial reservoir index (LA RI), ratio of active-to-total left atrial area change (active/total LA AC), late-diastolic LV wall motion velocity at the time of atrial contraction (A_m_), early-diastolic LV wall motion velocity during the phase of rapid ventricular filling (E_m_), ratio of E_m_-to-A_m_ (E_m_/A_m_), left ventricular diameter at end-diastole (LVID_d_), left ventricular volume at end-diastole (LVIV_d_), relative LV wall thickness at end-diastole (RWT_d_), left ventricular fractional shortening (LV FS), left ventricular ejection fraction (LV EF), stroke volume (SV), cardiac output (CO), pre-ejection period (PEP_m_), ejection time (ET_m_), index of myocardial performance (IMP_m_), and wall motion velocity during LV ejection (S_m_). Measurements of chamber dimensions were corrected for differences in body weight (BWT) according to the principles of allometric scaling [41,42,44,45]. Specifically, the measurements of LA and LV dimensions were normalized to a BWT of 500 kg using the following equations:Chamber diameter [500] (cm) = Measured chamber diameter/BWT^1/3^ × 500^1/3^Chamber area [500] (cm^2^) = Measured chamber area/BWT^2/3^ × 500^2/3^Chamber volume [500] (mL) = Measured chamber volume/BWT × 500

To enable direct comparison with heart rate-corrected Audicor^®^ variables, PEP_m_ and ET_m_ were corrected for heart rate, with RR being the electrocardiographic RR interval of the corresponding cardiac cycle, using the following equations:PEP_m-c_ (%) = PEP_m_/RR × 100ET_m-c_ (%) = ET_m_/RR × 100

### 2.5. Audicor^®^ Data Recordings

Data recordings were obtained by multiple operators at the Vetsuisse Faculty, University of Zurich, and at the Faculty of Veterinary Medicine, Ghent University, with the Audicor^®^ Dx Patch device (ApoDx Technologies, Taipei, Taiwan) using a previously described method [40]. In brief, the Dx Patch device was placed into a padded holding device and attached to the left side of the thorax using a surcingle. It was positioned between the 5th and 7th intercostal space, in a vertical orientation. Dry ECG electrodes with an Ag/AgCl core (Cognionics Inc., San Diego, CA, USA), wetting of the fur before placement, and tightening of the surcingle ensured a good electrical contact. Connection of the device to a laptop computer was obtained via a proprietary Wi-Fi-type low-power wireless connection. After starting the recording, a good quality, stable ECG and phonocardiogram were visually confirmed on a 10 s snapshot recording. After confirmation of sufficient quality, the wireless connection was discontinued, and the device continued recording to its internal data memory. Three continuous overnight recordings (AF day −1, NSR day 1, NSR day ≥ 2) were conducted in each horse in a standardized way. During the overnight recording, the horses were left undisturbed in the clinic stables. Procedures such as clinical examinations, feeding, and cleaning were continued as usual. After the conclusion of recordings, the raw data files were transferred via USB connection from the Dx Patch device onto a laptop computer.

### 2.6. Audicor^®^ Data Processing and Analysis

A detailed explanation of the diagnostic algorithm used to process and analyze the ECG data in relation to the heart sound data can be found elsewhere [34,40]. In short, the software algorithm incorporates wavelet-based signal processing techniques and time-frequency analysis for the evaluation of raw data [34] (Appendix A). According to information provided by the manufacturer (Inovise Medical Inc. (Beaverton, OR, USA), personal communication), comparison of the filtered signals from dogs, pigs, horses, and other animals to those of humans does not show any fundamental differences concerning their frequency content and the placement of the relevant fiducial points. However, no experimental data are currently available to validate these cross-species comparisons in horses specifically. Importantly, S4 is not detected or quantified when it occurs in isolation and lacks association with a QRS-T complex (e.g., with a second-degree AV block or in the context of atrial arrhythmia with AV blocks), representing a limitation of the current algorithm.

The data processing and analyses for this study were performed by a single operator (MS) using proprietary analysis software (CA300, Inovise Medical, Inc.) Manual page-by-page verification and correction of the ECG was performed using the scanning function of the program. This step was necessary because the automated detection is based on a human complex morphology algorithm, and it showed a tendency to misinterpret the equine T and P wave as a QRS complex. Artifact detection was performed by the automated proprietary algorithm and by visual control.

For each overnight recording, five consecutive, good-quality, analyzable 10 s snapshot analyses were generated. A snapshot refers to a continuous 10 s segment of the recording during which all heartbeats are automatically detected and processed by the proprietary software. Snapshots were selected based on ECG signal quality within the time frame of 8:00 p.m. and 9:00 p.m. If insufficient diagnostic snapshots were available in this time frame due to suboptimal recording quality, the next best five snapshots from the recording were selected. The following variables were generated for each snapshot analysis: HR, EMAT, heart rate-corrected EMAT (EMATc), left ventricular systolic time (LVST), heart rate-corrected LVST (LVSTc), power (as a function of intensity and persistence) of S3, power of S4, and systolic dysfunction index (SDI, as a function of QRS duration, QT interval, EMATc, and S3) [40] (Appendix A). Power is calculated by the software on a relative dimensionless scale from 1 to 10, reflecting both signal intensity and persistence. Importantly, values below 5 for S3 and S4 are considered artefactual and do not correspond to true heart sounds; this threshold was set by the software to ensure optimal accuracy compared to visual overread by phonocardiography experts and to define the actual presence of heart sounds in people (Inovise Medical Inc., personal communication) [35]. It should be noted that these detection algorithms and thresholds were originally developed and validated for human cardiac data and were applied here without species-specific adaptation. To enhance measurement reproducibility, for EMAT, EMATc, LVST, LVSTc, and SDI, the median value across the five selected snapshots was used for further analyses. This approach was chosen based on pilot data from a repeatability study (Appendix A) [46,47], which demonstrated improved reproducibility compared to relying on a single snapshot. For the power of S3 and S4, where presence may vary across snapshots, the maximum value across five consecutive snapshots was selected for further analyses.

Each overnight recording was also run through the automated acoustic cardiography analysis, which analyzed a fixed time window from 10:00 pm to 04:00 am (maximum analysis period: 6 h) and required at least 4 h of artifact-free ECG and phonocardiogram recordings to generate the so called “cardiac findings report” [40] (Appendix A). The variables reported as “cardiac findings” were averaged by the software over the whole analyzed time window and included the following: HR, QRS duration (QRS), heart rate-corrected QT interval (QTc), EMAT, EMATc, LVST, and LVSTc. Additionally, “cardiac findings” also included the percentage of 10 sec segments in the analysis period, in which power of S3 and S4 was ≥5 (on a scale of 0–10), EMAT was ≥15%, and SDI was ≥5 and ≥7.5 (on a scale of 0–10). As with the snapshots, for S3 and S4, the ≥5 threshold is set to provide optimal accuracy compared to visual overread by phonocardiography experts and to define the actual presence of heart sounds in people (Inovise Medical Inc., personal communication) [35]. In case of EMAT (≥15%) and SDI (≥5 and ≥7.5), these cut-off values indicate LV systolic dysfunction in people [31,48,49,50]. SDI ≥ 5 correlates to an ejection fraction (EF) <50% and SDI ≥ 7.5 correlates to an EF < 35% and high left ventricular end-diastolic pressure (LVEDP) [51]. Hence, the respective cut-off values used by the analysis software were those established by the manufacturer and approved by the FDA to guide diagnosis in human patients (Inovise Medical Inc., personal communication); owing to the proprietary system algorithms, they could not be adapted for the purpose of this study.

Table 1 provides an overview of the Audicor^®^ variables and their corresponding echocardiographic measurements, including a brief rationale for their comparison. Further details can be found elsewhere [40].

### 2.7. Statistical Analysis

All statistical and graphical analyses were performed by commercially available software: Microsoft Excel for Microsoft 365 (Microsoft Corporation, Redmond, WA, USA), GraphPad Prism for Windows, version 9.0.0 (GraphPad Software, San Diego, CA, USA), SigmaPlot for Windows, version 12 (Systat Software Inc., San Jose, CA, USA), and MedCalc for Windows, version 19.2.1 (MedCalc Software Ltd., Ostend, Belgium). The level of significance was set at *p* < 0.05.

For time points 2–7 d after conversion, the measurements obtained on different days were compared using a one-way analysis of variance (ANOVA). The results did not show any significant differences between time points (F test, *p* = 0.073 to 0.965). Therefore, recordings from days 2 to 7 were pooled as one time point for subsequent analyses (NSR day ≥ 2).

To investigate the effect of treatment (TVEC) on echocardiographic measurements of LA and LV size and function and on Audicor^®^ variables, all variables were compared between the three time points (AF day −1, NSR day 1, NSR day ≥ 2) using a repeated-measure ANOVA with Tukey’s multiple comparisons test. For the variables not measurable on AF day −1 (i.e., active LA FAC, active/total LA AC, Am, Em/Am), a paired *t*-test was performed to detect differences between NSR day 1 and NSR day ≥ 2. Homogeneity of variances was assessed by graphical display of the data, and validity of the normality assumption was confirmed by assessment of normal probability plots of residuals. Summary statistics were provided using mean and standard deviation (SD). The differences between time points were reported as the difference of means (d_means_) and the 95% confidence interval (95% CI) of the difference of means. Bonferroni correction was performed to adjust for the family-wise error rate within sets of multiple echocardiographic variables describing LA and LV size and function and Audicor^®^ variables. To assess the association between snapshot Audicor^®^ variables and echocardiographic measurements, multiple linear regression was performed; repeated measures over time in the same horse were accounted for using dummy variables (effects coding). Appropriateness of the linear model was assessed by graphical display of the data and assessment of normal probability plots of the residuals. Agreement between comparable TDI variables and Audicor^®^ variables was evaluated using Bland–Altman analyses.

## 3. Results

### 3.1. Echocardiography

Analyzable echocardiograms were obtained in 22/28 (79%) of horses at time point AF day −1, in 21/28 (75%) of horses at time point NSR day 1, and in 17/28 (61%) of horses at time point NSR day ≥ 2. Non-analyzable echocardiograms were primarily attributable to the multicenter study design, which made consistent optimization of image quality across centers challenging.

Table 2 summarizes the comparison of HR and echocardiographic variables of LA size and function between the three time points. Heart rate significantly decreased; LAD_llx-max_ (500), LAA_max_ (500), active LA FAC, LA RI, active/total LA AC, and A_m_ (measured by cTDI) significantly increased; and E_m_/A_m_ (measured by cTDI) significantly decreased after conversion. After Bonferroni correction to adjust for the family-wise error rate, only changes in HR, active LA FAC, LA RI, active/total LA AC, and A_m_ (measured by cTDI) remained significant.

Table 3 summarizes the comparison of echocardiographic variables of LV size and function between the three time points. The following indices were significantly altered after conversion: LVIV_d_ (500) significantly increased at time point NSR day ≥ 2 compared to AF day −1; SV significantly increased at time point NSR day ≥ 2 compared to AF day −1 and NSR day 1; CO significantly decreased on NSR day 1 compared to AF day −1 and significantly increased on NSR day ≥ 2 compared to NSR day 1; PEP_m_ (measured by cTDI) significantly increased; PEP_m-c_ (measured by PW TDI) significantly decreased at time point NSR day ≥ 2 compared to AF day −1; PEP_m-c_ (measured by cTDI) and ET_m-c_ (measured by PW TDI and cTDI) significantly decreased at time points NSR day 1 and NSR day ≥ 2 compared to AF day −1. After Bonferroni correction to adjust for the family-wise error rate, only changes in SV and in PEP_m-c_ and ET_m-c_ measured by PW TDI and cTDI remained significant.

### 3.2. Audicor^®^

Audicor^®^ recordings were attempted in all horses at all time points, but data quality was not consistent enough to obtain complete datasets in all horses. Motion artifacts, device failure, insufficient contact of the electrodes, and lack of recorded data in the fixed analysis time window (10:00 p.m.–04:00 a.m.) for “cardiac findings” reports were among the reasons for incomplete datasets.

At time point AF day −1, Audicor^®^ raw data recordings allowed extraction of “snapshots” in 22/28 (79%) and calculation of overnight “cardiac findings” reports in 12/28 (43%) of horses. There were missing data because of device failure in six horses, and there were insufficient data for the overnight analysis because of the length or timing of the recording (i.e., no or insufficient data between 10:00 p.m. and 04:00 a.m.) in the other 10 horses. The mean duration of recordings was 10:24 h (SD 5:15 h, range 0:17–17:25 h). The mean duration of analyzed recordings for the “cardiac findings” reports was 5:51 h (SD 00:31 h, range 4:07–6:00 h).

At time point NSR day 1, Audicor^®^ raw data recordings allowed extraction of “snapshots” in 21/28 (75%) and calculation of overnight “cardiac findings” reports in 14/28 (50%) of horses. There were missing data because of device failure in six horses, and there were insufficient data for the overnight analysis because of the length or timing of the recording (i.e., no or insufficient data between 10:00 pm and 04:00 am) in the other eight horses. The mean duration of recordings was 13:40 h (SD 6:15 h, range 0:14–23:32 h). The mean duration of analyzed recordings for the “cardiac findings” reports was 6:00 h (SD 00:01 h, range 5:58–6:00 h).

At time point NSR day ≥ 2, Audicor^®^ raw data recordings allowed extraction of “snapshots” in 21/28 (75%) and calculation of overnight “cardiac findings” reports in 13/28 (46%) of horses. There were missing data because of device failure in seven horses, and there were insufficient data for the overnight analysis because of the length or timing of the recording (i.e., no or insufficient data between 10:00 p.m. and 04:00 a.m.) in the other eight horses. The mean duration of recordings was 13:31 h (SD 05:37 h, range 01:43–20:00 h). The mean duration of analyzed recordings for the “cardiac findings” reports was 05:58 h (SD 00:06 h, range 05:38–06:00 h).

Table 4 summarizes the comparison of “snapshot” analyses between the three time points. The power of S4 was below 5 in all analyzed horses. Several variables significantly changed after treatment: HR, EMATc, LVSTc significantly decreased; LVST and power of S3 significantly increased. After Bonferroni correction to adjust for the family-wise error rate, all changes except the power of S3 remained significant.

Table 5 shows the comparison of “cardiac findings” analyses between the three time points. The power of S4 was below 5 in all analyzed horses, resulting in S4 (≥5) equaling zero. Therefore, it was not included in the statistics. Only LVSTc was significantly altered after treatment; however, after Bonferroni correction to adjust for family-wise error rate, it did not remain significant.

### 3.3. Association Between Audicor^®^ and Echocardiography

The associations between Audicor^®^ variables from “snapshot” analyses and echocardiographic indices are summarized in Figure 1, Figure 2 and Figure 3. Linear regression analyses indicated that EMAT and PEPm, EMATc and PEPm-c, and LVSTc and ETm-c were significantly related, with the adjusted R^2^ ranging between 0.383 and 0.55 for all relationships (Figure 1). No significant relationship was found for LVST and ETm and EMAT/LVST and PEPm/ETm. The mean biases and limits of agreement between corresponding TDI vs. Audicor^®^ variables, expressed as ratios, are shown in Figure 2. Linear regression analyses further indicated that EMAT, EMATc, LVST, LVSTc, and EMAT/LVST were significantly related to LVIVd (500) (adjusted R^2^ ranging between 0.548 and 0.631 for all relationships) and to SV (adjusted R^2^ ranging between 0.399 and 0.536 for all relationships) (Figure 3).

## 4. Discussion

This is the first study aiming to provide proof of concept for the use of acoustic cardiography to assess LA and LV function in the context of AF in horses.

To compare Audicor^®^ analyses against an established diagnostic method, echocardiographic examinations were performed as a reference to assess LA and LV size and function 1 day before and 1 day and 2–7 days after cardioversion. The echocardiographic studies confirmed previously reported findings of atrial mechanical dysfunction after conversion of AF to NSR and its partial recovery in the days following conversion [9,10,11,13,17,24,52]. Both LA contractile and reservoir function were decreased 1 day after conversion, indicating atrial stunning [9]. Changes in active LA FAC, LA RI, active/total LA AC, and Am (measured by color TDI) indicated an improvement in LA mechanical function from day 1 to 2–7 days after conversion, whilst still remaining below reference values [10,17,41].

In terms of LV function, a significant increase in SV and decrease in HR were observed in the study sample after conversion of AF to NSR, corroborating previous findings of AF’s negative impact on ventricular function and improvement thereof after treatment [6,7,53]. It is likely that the decrease in ventricular performance with AF is concomitant with the atrial mechanical dysfunction, reducing end-diastolic ventricular filling [54], as supported by the lower LVIV_d_ (500) before compared to the days after conversion of AF to NSR. To various extents, an increase in HR related to increased AV nodal conduction is seen in AF at rest, compensating for the reduced SV and maintaining CO [7,13,55]. It is critical to keep in mind that echocardiographic indices of systolic ventricular function are not a direct reflection of contractility, but are impacted by other factors such as preload, afterload, HR, and rhythm [21].

Previous studies have demonstrated changes in systolic time intervals in horses with cardiac disease [25,56,57,58]. In this study, TDI-derived systolic time intervals were not significantly altered after cardioversion. However, rate-corrected PEP_m-c_ and ET_m-c_ decreased significantly after conversion, indicating that lengthening of PEP_m_ and ET_m_ after cardioversion (which was not found to be statistically significant) was disproportional to lengthening of cycle length related to the significant decrease in HR. These rate-corrected systolic time intervals were calculated to enable direct comparison with the respective Audicor^®^ variables. Overall, these findings could further indicate improved LV systolic function after conversion of AF to NSR.

This study demonstrates the feasibility of Audicor^®^ recordings in a clinical setting. However, it was more difficult to acquire analyzable overnight recordings than was expected, given results from a previous experimental study [40]. No significant changes were found in “cardiac findings” variables after conversion. In “snapshot” variables, conversion of AF to NSR was associated with a significant decrease in EMATc and LVSTc as well as a significant increase in LVST. Hence, the “snapshot” analyses proved to be clinically more suitable for use in clinical settings than the analyses that were based on overnight recordings. This could be in part due to the low number of recordings that were sufficient in length and quality to provide the necessary base for the automatic “cardiac findings” analyses.

The EMAT reflects the time required for the LV after electrical activation to generate enough force to close the mitral valve and open the aortic valve. It is influenced by the rate at which sufficient left ventricular pressure develops [34] and has been strongly correlated to the maximum rate of systolic LV pressure rise (LV dP/dt_max_): A prolonged EMAT was associated with reduced LV dP/dt_max_ in people with LV systolic dysfunction [59,60], in both normal sinus rhythm [60] and AF [51]. Furthermore, a link between EMAT and EMATc, respectively, and left ventricular contractility has been proven, as impaired contractility was associated with abnormal EMATc (≥15%) [31,60,61]. Lastly, EMAT has been correlated with LV ejection fraction (LV EF) by echocardiography [32] and angiocardiography [59] and can be used to detect impaired LV systolic function in humans [59,60,62,63]. In the present study, EMATc significantly decreased after conversion, which likely indicates improved ventricular systolic function. This decrease in EMATc can at least partly be explained by the improved atrial function and thus improved LV preload. This effect of AF conversion on EMATc has also been shown in people [38]. However, for this equine study sample, EMATc during AF was still within reference range for horses in NSR [40] and below the human cut-off value (≥15%) for systolic dysfunction [31,48,49,50].

The LVST reflects the time between aortic valve opening and aortic valve closure, hence the time during which the LV is able to produce enough pressure to keep open the aortic valve during systole [59]. It has been related to angiographic LV EF and, thus, systolic function in people and is shortened in patients with LV systolic dysfunction [59]. In this study, LVST significantly increased after conversion of AF to NSR, indicating improved LV systolic function. These changes have also been shown in human patients [38] and are likely linked to improved LV preload because of the decrease in HR and improvement of atrial booster pump function after cardioversion. The LVSTc decreased after conversion, indicating that with decreasing HR, ventricular diastole was disproportionately extended compared to the prolonged LVST. This would also imply the potential for an overall improvement in cardiac function, as a shortened systole-to-diastole ratio improves coronary perfusion. These overall findings confirm the hypothesis that LV function improves after cardioversion.

Both S3 and S4 are considered pathologic in people, and their presence indicates impaired LV function [64]. The S3 is the result of cardiohemic vibrations caused by a sudden deceleration of ventricular inflow in early diastole, while S4 is the result of cardiohemic vibrations caused by a quick deceleration of ventricular inflow initiated by atrial contraction [64]. The use of S3 power as an indicator of LV dysfunction has been extensively investigated in human medicine. A strong correlation between S3 and LVEDP has been observed in patients with LV systolic dysfunction, where those values share a positive linear relationship [59,64]. A previous study in humans showed an increase in both S3 and S4 power with paroxysmal AF [38]. The S3 power has been used as a predictor for AF recurrence after electrical cardioversion [37] and for LV failure [63]. An increase in S4 power has been associated with LV stiffness and increased LVEDP [65], indicating impaired diastolic function [66].

In horses, however, both diastolic heart sounds, S3 and S4, are considered physiologic in most cases [39]. In a previous study, both S3 and S4 were detected using Audicor^®^ in healthy horses, with a prevalence of 0.2–19.7% and 0.1–13.0%, respectively, in overnight recordings [40]. Based on this study, Audicor^®^ quantification of S3 and S4 does not seem to be of clinical value to assess LA and LV function in the context of AF cardioversion. Neither of the diastolic heart sounds showed a significant change after conversion. A more intense S3 during AF, as seen in previous studies [43,67], was not detectable in the present study in horses, and S4 was not detectable above the cut-off of power ≥5 in any recording at any time point. The cut-off of power ≥5 is used in humans to define the actual presence of the heart sounds [35], while its relevance in horses remains unclear, and it is possible that this cut-off is not suitable for horses. Another possible explanation for the lack of S4 in this study sample is the fact that it is associated with atrial contraction [39]. Even though a significant increase in LA mechanical function was detectable in echocardiography, indices associated with LA contraction were still below the reference range after conversion. Therefore, it could be argued that atrial contraction after conversion improved but was still too weak to produce a detectable S4. Furthermore, the missing of S4 in post-conversion recordings could also be explained by the diagnostic algorithm, which neither detects nor quantifies S4 in context with AV blocks. As S4 is most pronounced in horses with slow heart rates and long PQ intervals [43,68,69], this could lead to a reduced number of S4 detected. Finally, placement of the Audicor^®^ device might not have been optimal for quantification of S3 and S4 since it was determined by the placement of the surcingle and, therefore, not optimized for recording of heart sounds.

This study indicates strong agreement between Audicor^®^ variables of systolic function and corresponding echocardiographic variables. Substantial agreement was observed between EMAT and PEP_m_ and between EMATc and PEP_m-c_. EMAT corresponds to PEP_m_, but is not identical to it, as EMAT does not contain the isovolumic contraction like PEP_m_ [59]. Similarly, substantial agreement was found between LVSTc and ET_m-c_. Again, these two values correspond; however, LVSTc is slightly longer, as it contains the isovolumic contraction [36,59]. Lastly, there was also a substantial association between Audicor^®^ variables of systolic function (EMAT, EMATc, LVST, LVSTc, EMAT/LVST) and echocardiographic variables of systolic function (SV) and preload (LVIV_d_ (500)).

In previous studies in people, Audicor^®^ performed well in detecting cardiac function compared to echocardiography [30,32,70]. In some cases, it even outperformed echocardiography in detecting LV systolic and diastolic dysfunction [32,38,60]: for example, in a study investigating LV function in human patients suffering from paroxysmal AF with preserved LVEF, Audicor^®^ was able to detect subclinical LV systolic and diastolic dysfunction, while it remained undetected in echocardiography [38]. In human diagnostic cardiology, Audicor^®^ offers some advantages compared to echocardiography: it is operator-independent, the position of the device is pre-defined, no expertise is needed to perform the recordings, and it can be obtained from a wide variety of positions, such as in a supine, laterally recumbent, or upright position [29]. However, in horses, some of these advantages dissipate, as manual read-through by an expert is required due to the human nature of the algorithm. We could not confirm the hypothesis that S4 would correlate with echocardiographic variables of LA mechanical function. Nevertheless, the hypothesis that certain Audicor^®^ variables correlate well with echocardiographic variables to assess LV systolic function was confirmed, as similar changes were seen in corresponding variables after conversion of AF to NSR.

The main limitation of this study is that the Audicor^®^ device and its proprietary signal processing algorithms were originally developed and validated for use in humans. Detection thresholds, such as the cut-off of 5 for S3 and S4 power, were derived from human validation studies and may not directly translate to horses. In particular, low-frequency components of equine heart sounds may require dedicated research to establish appropriate detection thresholds. Similarly, SDI, which was developed for use in human patients, was included in this study to allow initial exploration of its behavior in horses. However, its diagnostic value in the equine setting remains uncertain and warrants further investigation. The lack of equine-specific algorithm development limits the generalizability of the findings and may have affected the accuracy and sensitivity of heart sound detection and systolic time interval measurements. The application of human-derived algorithms also made the preparation of the recordings for analysis time-intensive, as manual page-by-page verification and correction of the ECG were necessary to ensure correct analysis. Furthermore, overnight recordings proved difficult to obtain in a clinical setting, as the use of the device on horses caused substantial motion artifacts, leading to a substantial loss of data. However, “snapshot” recordings were feasible to obtain efficiently in a clinical setting and were easily analyzable. Finally, as mentioned above, the position of the Audicor^®^ device was not optimized for the points of maximum intensity of the heart sounds in individual horses.

Further studies in other settings are needed to establish Audicor^®^ as a potentially useful adjunctive to echocardiography in horses with cardiac disease. While this study focused on comparative assessments, future mechanistic investigations evaluating the relationship between heart sounds and underlying hemodynamic events will be essential to provide the physiological basis for species-specific algorithm development. The generation of equine-adapted algorithms may ultimately improve diagnostic accuracy and optimize the clinical utility of acoustic cardiography in this species. In addition, future studies will be conducted using the next generation Audicor^®^ device, allowing optimized positioning and recording location for “snapshot” recordings, possibly improving data quality for detection of heart sounds, derivation of systolic time intervals, and quantification of heart sound intensity.

## 5. Conclusions

This study provided proof of concept for the use of Audicor^®^ in a clinical setting in the context of equine AF. “Snapshot” recordings proved to be feasible to obtain in a clinical setting, and the analyses provided additional information on LV systolic function, reflecting changes seen in echocardiography. However, the Audicor^®^ device under investigation does not seem to be clinically useful as a tool to directly assess LA mechanical function after conversion of AF to NSR, at least not within the time frame up to 7 days after successful cardioversion.

## Figures and Tables

**Figure 1 animals-15-01993-f001:**
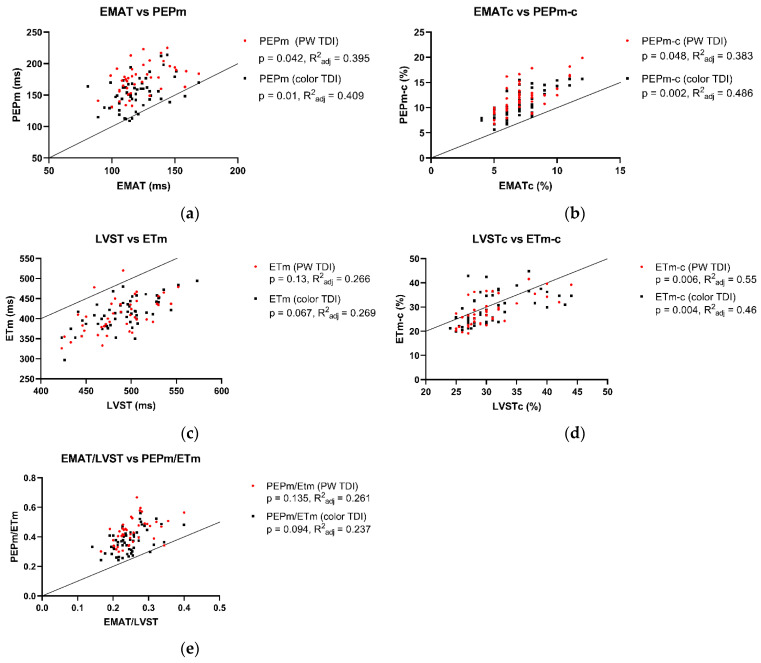
Association between Audicor^®^ variables from “snapshot” analyses and corresponding tissue Doppler imaging (TDI) variables. The black dots indicate color TDI measurements, and the red dots represent pulsed wave TDI measurements. (**a**–**e**): Linear regression analyses of association between Audicor^®^ variables and echocardiographic variables. Solid lines represent lines of identity. P, *p* value of linear regression statistics; R^2^_adj_, adjusted coefficient of determination; PW TDI, pulsed wave TDI; EMAT, electromechanical activation time; EMATc, heart rate-corrected electromechanical activation time; LVST, left ventricular systolic time; LVSTc, heart rate-corrected left ventricular systolic time; EMAT/LVST, ratio of EMAT-to-LVST; PEP_m_, pre-ejection period; PEP_m-c_, heart rate-corrected pre-ejection period; ET_m_, ejection time; ET_m-c_, heart rate-corrected ejection time; PEP_m_/ET_m_, ratio of PEP_m_-to-ET_m_. For detailed explanation of echocardiographic indices and Audicor^®^ variables, see Appendix A.

**Figure 2 animals-15-01993-f002:**
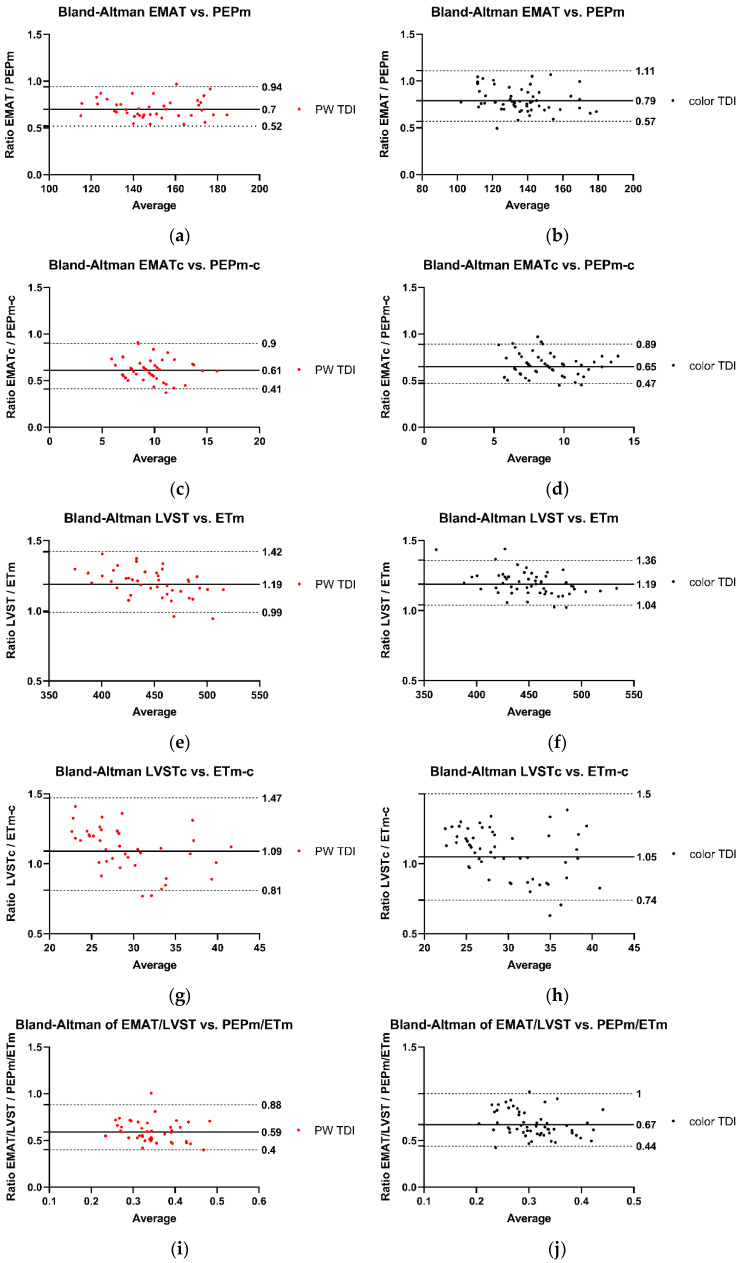
Agreement between Audicor^®^ variables from “snapshot” analyses and echocardiographic variables. (**a**–**j**): Bland–Altman analyses. The solid lines represent the mean bias, the dotted lines illustrate the lower and the upper 95% limit of agreement. The black dots indicate measurements performed with color TDI, and the red dots represent measurements performed with pulsed wave TDI. PW TDI, pulsed wave TDI; EMAT, electromechanical activation time; EMATc, heart rate-corrected electromechanical activation time; LVST, left ventricular systolic time; LVSTc, heart rate-corrected left ventricular systolic time; EMAT/LVST, ratio of EMAT-to-LVST; PEP_m_, pre-ejection period; PEP_m-c_, heart rate-corrected pre-ejection period; ET_m_, ejection time; ET_m-c_, heart rate-corrected ejection time; PEP_m_/ET_m_, ratio of PEP_m_-to-ET_m_. For detailed explanation of echocardiographic indices and Audicor^®^ variables, see Appendix A.

**Figure 3 animals-15-01993-f003:**
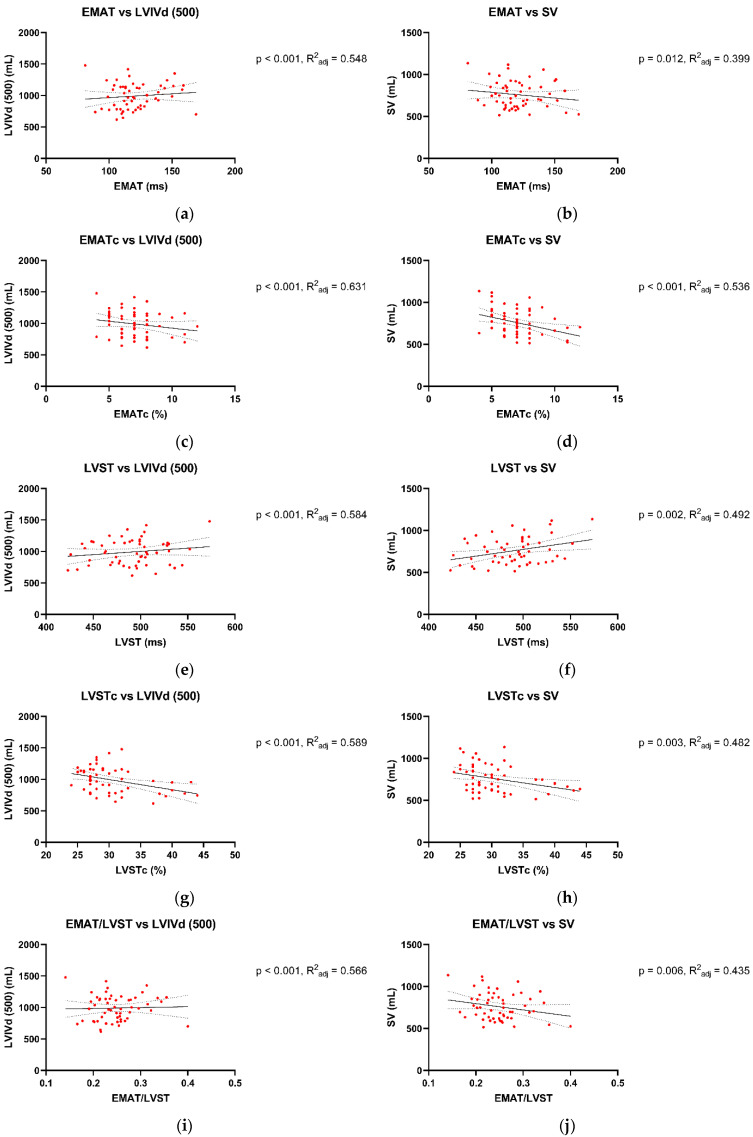
Association between Audicor^®^ variables from “snapshot” analyses and 2D echocardiographic variables. (**a**–**j**): Linear regression analyses of association between Audicor^®^ variables and echocardiographic variables. The solid lines represent the regression line, and the dotted lines illustrate the 95% confidence intervals of the regression lines. P, *p* value of linear regression statistics; R^2^_adj_, coefficient of determination; EMAT, electromechanical activation time; EMATc, heart rate-corrected electromechanical activation time; LVST, left ventricular systolic time; LVSTc, heart rate-corrected left ventricular systolic time; EMAT/LVST, ratio of EMAT-to-LVST; LVIV_d_, left ventricular volume at end-diastole; SV, stroke volume. For detailed explanation of echocardiographic indices and Audicor^®^ variables, see Appendix A.

**Table 1 animals-15-01993-t001:** Comparative table of Audicor^®^ variables and echocardiographic variables.

Audicor^®^ Variable	Echocardiographic Variable	Rationale for Comparison
EMAT	PEP_m_	Both represent early systolic time intervals related to electromechanical activation; compared to assess agreement, acknowledging that EMAT is slightly shorter, as it excludes isovolumic contraction.
	LVIV_d_ (500)	LVIV_d_ reflects preload, which may influence early systolic time intervals such as EMAT; included to explore preload effects on EMAT.
	SV	SV represents a global index of systolic function: compared to EMAT to evaluate their agreement as systolic function markers.
LVST	ET_m_	Both reflect systolic ejection period; compared to assess agreement, acknowledging that LVST is slightly longer, as it includes isovolumic contraction.
	LVIV_d_ (500)	LVIV_d_ reflects preload, which may influence ejection time intervals such as LVST; included to explore preload effects on LVST.
	SV	SV represents a global index of systolic function; compared to LVST to assess their agreement as systolic function markers.

For detailed explanation of variables, see Appendix A. EMAT, electromechanical activation time; PEP_m_, pre-ejection period; LVST, left ventricular systolic time; LVIV_d_, left ventricular volume at end-diastole; SV, stroke volume; ET_m_, ejection time.

**Table 2 animals-15-01993-t002:** Summary statistics for heart rate and echocardiographic variables of left atrial size and function, by time point.

Variable	Unit	AF Day −1	NSR Day 1	NSR ≥ 2	
		Mean ± SD [n]	Mean ± SD [n]	Mean ± SD [n]	*p* Value (F-Test or *t*-Test)
			d_means_ (95% CI) #	d_means_ (95% CI) §	
				d_means_ (95% CI) $	
HR	min	52 ± 10 [20]	36 ± 6 [21]	34 ± 5 [16]	<0.0001 *
			−16 (−23 to −9)	−17 (−26 to −9)	
				−1 (−6 to 3)	
Variables of LA size					
LAD_max_ (500)	cm	12.2 ± 1.4 [22]	12.6 ± 1.3 [21]	12.7 ± 1.1 [17]	0.1297
			0.4 (−0.3 to 1.0)	0.5 (−0.1 to 1.1)	
				0.1 (−0.4 to 0.6)	
LAD_max_/LVID_d_		1.09 ± 0.13 [20]	1.08 ± 0.10 [21]	1.10 ± 0.09 [16]	0.5805
			−0.01 (−0.08 to 0.06)	0.01 (−0.08 to 0.10)	
				0.02 (−0.03 to 0.07)	
LAD_llx-max_ (500)	cm	12.8 ± 1.0 [21]	13.1 ± 1.0 [22]	13.3 ± 0.8 [17]	0.0135
			0.3 (−0.1 to 0.6)	0.5 (0.1 to 0.9)	
				0.3 (−0.2 to 0.7)	
LAA_max_ (500)	cm^2^	98.2 ± 10.4 [22]	103.4 ± 12.4 [21]	105.4 ± 9.2 [17]	0.0229
			5.2 (−1.0 to 11.5)	7.2 (2.4 to 11.9)	
				2.0 (−3.4 to 7.3)	
LA_sx_A_max_ (500)	cm^2^	115.4 ± 12.6 [20]	118.5 ± 13.8 [21]	119.8 ± 11.8 [17]	0.1468
			3.0 (−2.4 to 8.5)	4.4 (−2.4 to 11.3)	
				1.4 (−5.4 to 8.2)	
Variables of LA function					
active LA FAC	%	n/a	−11 ± 7 [15]	−2 ± 6 [15]	<0.0001 *
			n/a	n/a	
				9 (6 to 12)	
LA RI	%	24 ± 10 [22]	29 ± 9 [21]	38 ± 12 [17]	<0.0001 *
			5 (−0.4 to 11)	14 (8 to 20)	
				9 (3 to 15)	
Active/total LA AC		n/a	−0.36 ± 0.27 [14]	−0.05 ± 0.15 [14]	0.0001 *
			n/a	n/a	
				0.31 (0.19 to 0.44)	
A_m_ (PW TDI)	cm/s	n/a	4.7 ± 1.8 [10]	6.0 ± 2.4 [10]	0.1217
			n/a	n/a	
				1.3 (−0.4 to 3.0)	
A_m_ (cTDI)	cm/s	n/a	2.9 ± 1.0 [13]	4.8 ± 2.2 [13]	0.0027 *
			n/a	n/a	
				1.9 (0.8 to 3.1)	
E_m_/A_m_ (PW TDI)		n/a	6.1 ± 2.6 [10]	4.9 ± 1.7 [10]	0.1975
			n/a	n/a	
				−1.2 (−3.1 to 0.8)	
E_m_/A_m_ (cTDI)		n/a	9.0 ± 3.7 [13]	6.1 ± 2.5 [13]	0.0079
			n/a	n/a	
				−2.9 (−4.9 to −0.9)	

For detailed explanation of variables, see Appendix A. HR; heart rate; LAD_max_ and LAD_llx-max_, maximum left atrial diameters; LVID_d_, left ventricular diameter at end-diastole; LAA_max_ and LA_sx_A_max_, maximum left atrial areas; active LA FAC, left atrial active fractional area change; LA RI, left atrial reservoir index; active/total LA AC, ratio of active-to-total left atrial area change; A_m_, late-diastolic LV wall motion velocity at the time of atrial contraction; E_m_, early-diastolic LV wall motion velocity during the phase of rapid ventricular filling; E_m_/A_m_, ratio of Em-to-Am; PW TDI, pulsed-wave TDI; cTDI, color TDI. SD, standard deviation; d_means_ (95% CI), difference of means (95% confidence interval of difference of means). * Significant after Bonferroni correction for 10 variables of cardiac function (8 of LA size and function, 2 ratios) (*p* < 0.005). # NSR day 1—AF day −1. § NSR day ≥ 2—AF day −1. $ NSR day ≥ 2—NSR day 1.

**Table 3 animals-15-01993-t003:** Summary statistics for echocardiographic variables of left ventricular size and function, by time point.

Variable	Unit	AF Day −1	NSR Day 1	NSR ≥ 2	
		Mean ± SD [n]	Mean ± SD [n]	Mean ± SD [n]	*p* Value (F-Test or *t*-Test)
			d_means_ (95% CI) #	d_means_ (95% CI) §	
				d_means_ (95% CI) $	
Variables of LV size					
LVID_d_ (500)	cm	11.3 ± 1.3 [20]	11.7 ± 1.2 [21]	11.6 ± 1.0 [16]	0.2281
			0.5 (−0.2 to 1.1)	0.3 (−0.1 to 0.8)	
				−0.1 (−0.7 to 0.5)	
LVIV_d_ (500)	mL	928 ± 203 [19]	979 ± 191 [21]	1068 ± 201 [17]	0.0068
			51 (−60 to 162)	140 (54 to 226)	
				89 (−19 to 196)	
RWT_d_		0.500 ± 0.065 [20]	0.464 ± 0.068 [21]	0.454 ± 0.048 [16]	0.0617
			−0.036 (−0.085 to 0.013)	−0.046 (−0.083 to −0.009)	
				−0.010 (−0.054 to 0.035)	
Variables of LV function measured by 2D and M-mode echocardiography					
LV FS	%	36 ± 8 [20]	36 ± 9 [21]	41 ± 5 [16]	0.5055
			0 (−3 to 4)	5 (−1 to 10)	
				5 (−1 to 10)	
LV EF	%	65 ± 6 [19]	67 ± 7 [21]	71 ± 5 [17]	0.0866
			2 (−2 to 5)	6 (2 to 10)	
				4 (2 to 7)	
SV	mL	671 ± 103 [19]	729 ± 160 [21]	886 ± 144 [17]	0.0001 *
			58 (−37 to 152)	215 (122 to 307)	
				157 (67 to 248)	
CO	L	31.8 ± 5.4 [19]	26.4 ± 6.0 [21]	31.6 ± 6.3 [17]	0.0046
			−5.4 (−8.9 to −1.8)	−0.1 (−6.2 to 5.9)	
				5.2 (0.9 to 9.6)	
Variables of LV function measured with PW TDI					
PEP_m_	ms	168 ± 26 [15]	181 ± 24 [16]	176 ± 20 [12]	0.2119
			13 (−4 to 31)	9 (−22 to 40)	
				−5 (−17 to 8)	
PEP_m-c_	%	14 ± 3 [15]	11 ± 2 [16]	10 ± 2 [12]	0.0015 *
			−3 (−6 to 0.1)	−4 (−8 to −0.2)	
				−1 (−3 to 0.4)	
ET_m_	ms	400 ± 52 [15]	406 ± 45 [16]	427 ± 28 [12]	0.2415
			6 (−28 to 39)	27 (−32 to 85)	
				21 (−10 to 51)	
ET_m-c_	%	34 ± 5 [15]	25 ± 4 [16]	24 ± 3 [12]	<0.0001 *
			−9 (−14 to −5)	−10 (−15 to −5)	
				−1 (−3 to 1)	
PEP_m_/ET_m_		0.428 ± 0.101 [15]	0.454 ± 0.093 [16]	0.416 ± 0.056 [12]	0.4254
			0.026 (−0.051 to 0.103)	−0.013 (−0.139 to 0.113)	
				−0.039 (−0.092 to 0.015)	
IMP_m_		0.387 ± 0.120 [15]	0.451 ± 0.116 [16]	0.405 ± 0.087 [12]	0.2551
			0.064 (−0.020 to 0.148)	0.017 (−0.138 to 0.172)	
				−0.047 (−0.135 to 0.041)	
S_m_	cm/s	9.8 ± 1.9 [15]	9.3 ± 1.5 [16]	10.2 ± 1.6 [12]	0.2325
			−0.6 (−1.7 to 0.6)	0.4 (−1.5 to 2.3)	
				0.9 (−0.5 to 2.4)	
E_m_	cm/s	28.4 ± 6.3 [15]	26.4 ± 4.8 [16]	26.3 ± 4.8 [12]	0.1693
			−2.0 (−6.9 to 3.0)	−2.0 (−6.8 to 2.5)	
				−0.2 (−4.6 to 4.2)	
Variables of LV function measured with cTDI					
PEP_m_	ms	142 ± 21 [20]	157 ± 28 [21]	158 ± 21 [16]	0.0348
			14 (−1 to 29)	16 (−3 to 35)	
				2 (−13 to 17)	
PEP_m-c_	%	13 ± 2 [20]	10 ± 2 [21]	9 ± 2 [16]	<0.0001 *
			−3 (−5 to −1)	−4 (−6 to −2)	
				−1 (−2 to 1)	
ET_m_	msc	400 ± 43 [20]	417 ± 33 [21]	426 ± 37 [16]	0.1517
			17 (−14 to 48)	26 (−18 to 70)	
				9 (−15 to 32)	
ET_m-c_	%	36 ± 4 [20]	26 ± 4 [21]	24 ± 2 [16]	<0.0001 *
			−10 (−13 to −7)	−12 (−15 to −9)	
				−2 (−4 to 1)	
PEP_m_/ET_m_		0.362 ± 0.079 [20]	0.380 ± 0.086 [21]	0.376 ± 0.071 [16]	0.6519
			0.018 (−0.042 to 0.077)	0.014 (−0.067 to 0.096)	
				−0.003 (−0.059 to 0.05)	
IMP_m_		0.421 ± 0.088 [20]	0.431 ± 0.095 [21]	0.423 ± 0.080 [16]	0.8525
			0.010 (−0.045 to 0.065)	0.002 (−0.084 to 0.088)	
				−0.008 (−0.078 to 0.062)	
S_m_	cm/s	7.9 ± 1.5 [20]	7.7 ± 1.5 [21]	8.5 ± 1.5 [16]	0.2158
			−0.2 (−1.5 to 0.7)	0.6 (−0.1 to 1.3)	
				0.8 (−0.02 to 1.5)	
E_m_	cm/s	25.5 ± 4.0 [20]	22.8 ± 4.9 [21]	24.6 ± 4.4 [16]	0.0679
			−2.7 (−5.6 to 0.3)	−0.8 (−3.6 to 2.0)	
				1.9 (−1.1 to 4.8)	

For detailed explanation of variables, see Appendix A. LVID_d_, left ventricular diameter at end-diastole; LVIV_d_, left ventricular volume at end-diastole; RWT_d_, relative LV wall thickness at end-diastole; LV FS, left ventricular fractional shortening; LV EF, left ventricular ejection fraction; SV, stroke volume; CO, cardiac output; PEP_m_, pre-ejection period; PEP_m-c_, heart rate-corrected pre-ejection period; ET_m_, ejection time; ET_m-c_, heart rate-corrected ejection time; PEP_m_/ET_m_, ratio of PEP_m_-to-ET_m_; IMP_m_, index of myocardial performance; S_m_, wall motion velocity during LV ejection; E_m_, early-diastolic LV wall motion velocity during the phase of rapid ventricular filling; PW TDI, pulsed-wave TDI; cTDI, color TDI. SD, standard deviation; d_means_ (95% CI), difference of means (95% confidence interval of difference of means). * Significant after Bonferroni correction for 17 variables of cardiac function (15 of LV size and function, 2 ratios) (*p* < 0.003). # NSR day 1—AF day −1. § NSR day ≥ 2—AF day −1. $ NSR day ≥ 2—NSR day 1.

**Table 4 animals-15-01993-t004:** Comparison of Audicor^®^ snapshot variables between the three time points.

Variable	Unit	AF Day −1	NSR Day 1	NSR ≥ 2	
		Mean ± SD [n]	Mean ± SD [n]	Mean ± SD [n]	*p* Value (F-Test or *t*-Test)
			d_means_ (95% CI) #	d_means_ (95% CI) §	
				d_means_ (95% CI) $	
HR	min	43 ± 6 [22]	34 ± 3 [21]	33 ± 4 [21]	<0.0001 *
			−9 (−12 to −6)	−10 (−13 to −7)	
				−1 (−3 to 1)	
EMAT	ms	124 ± 15 [22]	123 ± 18 [21]	119 ± 24 [21]	0.3852
			−1 (−10 to 8)	−5 (−16 to 6)	
				−4 (−18 to 10)	
EMATc	%	8 ± 2 [22]	7 ± 1 [21]	6 ± 2 [21]	<0.0001 *
			−2 (−3 to −1)	−2 (−3 to −1)	
				−0.4 (−2 to 1)	
LVST	ms	473 ± 26 [22]	498 ± 30 [21]	503 ± 32 [21]	0.0001 *
			25 (10 to 41)	31 (19 to 42)	
				5 (−12 to 23)	
LVSTc	%	35 ± 5 [22]	28 ± 2 [21]	27 ± 2 [21]	<0.0001 *
			−6 (−9 to −4)	−7 (−10 to −5)	
				−1 (−3 to 1)	
EMAT/LVST		0.264 ± 0.038 [22]	0.248 ± 0.047 [21]	0.239 ± 0.058 [21]	0.1112
			−0.016 (−0.041 to 0.010)	−0.025 (−0.049 to −0.002)	
				−0.01 (−0.047 to 0.028)	
S3 (°max)	1–10	5.2 ± 0.6 [22]	5.2 ± 1.3 [21]	6.1 ± 1.7 [21]	0.0167
			0.1 (−0.8 to 0.9)	1.0 (−0.1 to 2.0)	
				0.9 (−0.03 to 1.9)	
S4 (°max)	1–10	3.3 ± 0.5 [22]	2.9 ± 0.7 [21)	2.9 ± 0.9 [21]	0.2027
			−0.4 (−0.8 to 0.1)	−0.3 (−1.0 to 0.3)	
				0.03 (−0.5 to 0.6)	
SDI	1–10	3.2 ± 0.8 [21]	3.5 ± 1.2 [21)	3.4 ± 1.1 [20]	0.4265
			0.3 (−0.5 to 1.1)	0.3 (−0.2 to 0.7)	
				−0.1 (−0.8 to 0.7)	

For detailed explanation of variables, see Appendix A. HR, heart rate; EMAT, electromechanical activation time; EMATc, heart rate-corrected electromechanical activation time; LVST, left ventricular systolic time; LVSTc, heart rate-corrected left ventricular systolic time; EMAT/LVST, ratio of EMAT-to-LVST; S3, power of the third heart sound; S4, power of the fourth heart sound; SDI, systolic dysfunction index. SD, standard deviation; d_means_ (95% CI), difference of means (95% confidence interval of difference of means). * Significant after Bonferroni correction for 8 variables of cardiac function (excluding HR) (*p* < 0.006). # NSR day 1—AF day −1. § NSR day ≥ 2—AF day −1. $ NSR day ≥ 2—NSR day 1.

**Table 5 animals-15-01993-t005:** Comparison between Audicor^®^ cardiac findings variables between the three time points.

Variable	Unit	AF Day −1	NSR Day 1	NSR ≥ 2	
		Mean ± SD [n]	Mean ± SD [n]	Mean ± SD [n]	*p* Value (F-Test or *t*-Test)
			d_means_ (95% CI) #	d_means_ (95% CI) §	
				d_means_ (95% CI) $	
HR	min	37 ± 5 [12]	35 ± 4 [14]	34 ± 6 [13]	0.3914
			−1 (−6 to 4)	−2 (−9 to 5)	
				−1 (−4 to 2)	
QRS	ms	111 ± 8 [12]	114 ± 11 [14]	119 ± 14 [13]	0.1427
			3 (−4 to 10)	9 (−3 to 20)	
				6 (−5 to 16)	
QTc	%	366 ± 21 [12]	386 ± 23 [14]	373 ± 28 [13]	0.1214
			20 (−3 to 43)	7 (−30 to 43)	
				−14 (−46 to 19)	
EMAT	ms	128 ± 12 [12]	124 ± 23 [14]	118 ± 21 [13]	0.6873
			−4 (−20 to 12)	−10 (−26 to 7)	
				−6 (−19 to 8)	
EMATc	%	10 ± 2 [12]	10 ± 4 [14]	11 ± 4 [13]	0.4870
			−0.2 (−4 to 4)	0.3 (−3 to 3)	
				1 (−1 to 2)	
LVST	ms	439 ± 54 [12]	444 ± 62 [14]	412 ± 86 [13]	0.4967
			5 (−42 to 51)	−27 (−114 to 60)	
				−32 (−89 to 26)	
LVSTc	%	34 ± 2 [12]	31 ± 3 [14]	30 ± 4 [13]	0.0104
			−3 (−6 to −0.4)	−4 (−7 to −1)	
				−1 (−4 to 2)	
EMAT/LVST		0.296 ± 0.048 [12]	0.284 ± 0.060 [14]	0.304 ± 0.107 [13]	0.6474
			−0.013 (−0.063 to 0.038)	0.008 (−0.096 to 0.112)	
				0.021 (−0.069 to 0.111)	
S3 ≥ 5	%	10 ± 10 [12]	11 ± 13 [14]	10 ± 16 [13]	0.6265
			1 (−7 to 8)	−0.2 (−11 to 11)	
				−1 (−9 to 8)	
SDI ≥ 5	%	5 ± 7 [12]	8 ± 15 [14]	13 ± 13 [13]	0.1034
			4 (−12 to 19)	8 (−3 to 19)	
				4 (−5 to 13)	
EMATc ≥ 15	%	11 ± 13 [12]	18 ± 23 [14]	24 ± 23 [13]	0.1274
			7 (−20 to 34)	13 (−10 to 35)	
				6 (−5 to 16)	

For detailed explanation of variables, see Appendix A. HR, heart rate; QRS, QRS duration; QTc, rate-corrected QT interval; EMAT, electromechanical activation time; EMATc, heart rate-corrected electromechanical time; LVST, left ventricular systolic time; LVSTc, heart rate-corrected left ventricular systolic time; EMAT/LVST, ratio of EMAT-to-LVST; S3 (≥5), per cent detected 10 s segments in which the power of the third heart sound is greater than or equal the threshold value of 5; S4 (≥5), per cent detected 10 s segments in which the power of the fourth heart sound is greater than or equal the threshold value of 5; SDI (≥5), per cent detected 10 s segments in which the systolic dysfunction index is greater than or equal the threshold value of 5; EMATc (≥15%), per cent detected 10 s segments in which electromechanical activation is greater than or equal the threshold value of 15%. SD, standard deviation; d_means_ (95% CI), difference of means (95% confidence interval of difference of means). # NSR day 1—AF day −1. § NSR day ≥ 2—AF day −1. $ NSR day ≥ 2—NSR day 1.

## Data Availability

The data are available from the authors upon reasonable request.

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
