# Peer review of "Application of Acoustic Cardiography in Assessment of Cardiac Function in Horses with Atrial Fibrillation Before and After Cardioversion"

_animals, 2025, doi:10.3390/ani15131993_

Round 1
Reviewer 1 Report
Comments and Suggestions for Authors
The manuscript entitled “Quantification of cardiac function in horses with atrial fibrillation before and after cardioversion using ambulatory acoustic cardiography” investigate the use of Audicor® as a potentially useful adjunctive to echocardiography in horses with cardiac disease. It’s a well-structured manuscript, complete easy to follow and understand. Some minor correction should be reported in order to let this manuscript accepted for publication. Below some minor corrections are reported.
Keywords
Atrial fibrillation and Audicor® should be included
Materials and methods
Line 118 what do you mean with physical examination? Some additional information are required
Line 104-124 were the horses in the two sampling areas managed in the same way through the same environmental and managerial conditions?
Line 113-115 during the experimental phase were the horses in sporting activity? were they at rest? where were they kept? how long before and after the experiment were the horses not in training possibly?
Results and Discussion
Please revise figure 1 presentation (lines)
Author Response
|
Response to Reviewer 1 Comments |
||
|
1. Summary |
|
|
|
Thank you very much for taking the time to review this manuscript. Please find the detailed responses below and the corresponding revisions/corrections highlighted/in track changes in the re-submitted files.
|
||
|
2. Point-by-point response to Comments and Suggestions for Authors |
||
|
Comments 1: Atrial fibrillation and Audicor® should be included in the Keywords. Response 1: As atrial fibrillation is already included in the title of the manuscript, and indexing systems automatically include title words in search algorithms, we believe it is not necessary (nor requested by the journal) to repeat it in the keywords. However, we agree that including Audicor® as a keyword would enhance discoverability and have therefore added it to the list (page 1, line 47). Updated Keywords: “arrhythmia; phonocardiogram; echocardiography; cardiovascular; Audicor®” |
||
|
|
||
|
Comments 2: Line 118 what do you mean with physical examination? Some additional information are required Response 2: The term “physical examination” refers to a standard clinical examination of the horse commonly performed in equine practice. To clarify for broader readership, we have expanded the description in the manuscript to include the specific components assessed. This change can be found on page 2, line 123-127 of the revised manuscript. “All horses underwent a physical examination (demeanor and attitude, body condition score, skin and hair coat, jugular veins, peripheral pulse rate and quality, cardiac auscultation (heart rate (HR), rhythm, murmurs), respiratory rate and lung sounds, mucous membrane color and capillary refill time, gastrointestinal sounds, and rectal temperature), echocardiographic examination, and Audicor® examination at three time points: (1) AF day -1, one day before cardioversion; (2) NSR day 1, one day after cardioversion; and (3) NSR day ≥2, two to seven days after cardioversion.”
Comment 3: Line 104-124 were the horses in the two sampling areas managed in the same way through the same environmental and managerial conditions? Response 3: The horses were housed in similar hospital barn environments rather than their home environments, but at two different locations. This means there were some differences in environmental and managerial conditions, such as temperature, bedding, and feeding. While these differences exist, we do not believe they significantly influenced the cardiac function parameters measured in our study.
Comment 4: Line 113-115 during the experimental phase were the horses in sporting activity? were they at rest? where were they kept? how long before and after the experiment were the horses not in training possibly? Response 4: During the experimental phase, the horses were not engaged in sporting activity and were at rest. However, beyond this, training status and management were not standardized, and detailed data on the duration of rest or training cessation before and after the experiments were not available. We believe this variability reflects typical clinical conditions and is unlikely to have significantly influenced the cardiac function parameters assessed in our study.
Comment 5: Please revise figure 1 presentation (lines). Response 5: We have revised Figure 1 by removing all table borders to improve clarity and visual presentation. To ensure continuity and a consistent style throughout the manuscript, we applied the same revision to Figures 2 and 3. |
||
Reviewer 2 Report
Comments and Suggestions for Authors
Simples summary:
- Line 21: The term (Audicor) should be omitted, as previously described.
Introduction:
- Line 92/95/96: The sentence should be revised to employ a third-person verb form.
Discussion:
- Line 470: Verify the spelling thereof.
Conclusion:
- Line 604: It is recommended that this point be incorporated into the discussion section, as including it in the conclusion does not add significant value to the research:
Further studies in other settings are needed to establish Audicor® as a potentially useful adjunct to echocardiography in horses with heart disease. Future studies will also be conducted using the next-generation Audicor® device, allowing optimized positioning and recording location for 'instantaneous' recordings, potentially improving data quality for heart sound detection, derivation of systolic time intervals and quantification of heart sound intensity.
References:
References must be cited according to the model.
- Autor, AB; Autor, C.D. Título do artigo. Nome abreviado do periódico Ano, Volume, intervalo de páginas.
Author Response
|
Response to Reviewer 2 Comments |
||
|
1. Summary |
|
|
|
Thank you very much for taking the time to review this manuscript. Please find the detailed responses below and the corresponding revisions/corrections highlighted/in track changes in the re-submitted files
|
||
|
2. Point-by-point response to Comments and Suggestions for Authors |
||
|
Comment 1: Line 21: The term (Audicor) should be omitted, as previously described. Response 1: We have omitted the term Audicor® from Line 21 (page 0, line 23). Revised text: “This study aimed to evaluate the use of an acoustic cardiography monitor to assess cardiac mechanical function and its benefits as an adjunct to echocardiography in horses with AF before and after successful cardioversion to sinus rhythm.”
Comment 2: Line 92/95/96: The sentence should be revised to employ a third-person verb form. Response 2: The sentences have been revised to employ the correct third-person verb form (page 2, lines 94, 97, and 99). Revised text: “It was hypothesized that S4 would not be detectable by Audicor® during AF and that the strength of S4 would progressively increase on subsequent follow-up examinations after restoration of NSR, indicating recovery of LA mechanical function. Furthermore, it was postulated that the strength of S4 would correlate to echocardiographic variables of LA mechanical function. Lastly, it was hypothesized that Audicor® variables of LV function would improve after cardioversion, mirroring echocardiographic variables of LV function.”
Comment 3: Line 470: Verify the spelling thereof. Response 3: Thank you for the comment. We have verified the spelling of “thereof” and confirm that it is correct.
Comment 4: Line 604: It is recommended that this point be incorporated into the discussion section, as including it in the conclusion does not add significant value to the research: Further studies in other settings are needed to establish Audicor® as a potentially useful adjunct to echocardiography in horses with heart disease. Future studies will also be conducted using the next-generation Audicor® device, allowing optimized positioning and recording location for 'instantaneous' recordings, potentially improving data quality for heart sound detection, derivation of systolic time intervals and quantification of heart sound intensity. Response 4: We agree. The point regarding the need for further studies with Audicor® and the future use of the next-generation device has now been incorporated into the Discussion section(page 25, lines 786-798). |
||
Comment 5: References must be cited according to the model.
- Autor, AB; Autor, C.D. Título do artigo. Nome abreviado do periódico Ano, Volume, intervalo de páginas.
Response 5: We have reviewed and revised all references to ensure they conform to the required citation format.
Reviewer 3 Report
Comments and Suggestions for Authors
Animals 3619836-peer-review-v1
As succinctly stated in the Simple Summary, "This study aimed to evaluate the use of an acoustic monitor (Audicor®) to assess cardiac mechanical function and its benefits as an adjunct to echocardiography in horses with AF before and after successful cardioversion to sinus rhythm". The paper achieves this goal very effectively, while addressing, briefly, the potential benefits of using a technology such as this, once validated. in the absence of specialized referral facilities to provide guidance to supervising clinicians.
Conversely, there is a degree of emphasis on the echocardiographic measurements taken and their demonstration of chamber dysfunction in AF cases after cardioversion that exceeds that necessary to achieve the paper’s primary objective. As a result, the paper is long and far more complex than necessary. The issue of chamber dysfunction in AF has been very thoroughly researched and confirmed by several authors and the coherence of present findings with this existing literature needs no more than a statement to that effect. To this might be added confirmation that the tested device produces results useful in assessing patient progress. Ironically, the authors conclude that the device's performance fell short in this regard. I recommend greatly reducing the emphasis on changes in cardiac function in AF cases, simply reference the existing literature, and concentrate on addressing the evaluation of the equipment. Reference to citation [40] could address much of the need in this area.
The analysis performed of the data collected is extensive and very thorough. However, there is also a lot of missing raw data, which combined with the lack of detail on the experimental subjects and on the protocols followed in the involved institutions, makes it difficult to assess the quality and relevance of the information gathered. In some instances the reasons for the missing data are addressed, in others these explanations are missing. As an example, there appear to have been some ultrasound data that could not be analysed. The recommendation that the authors shorten the paper and concentrate on the evaluation of the Audicor® equipment might make this point of limited importance, but if not, these issues should perhaps be more fully addressed.
The data used are multicentre convenience data. This is a valid approach, though little information is presented on patent specifics. Greater detail is required on the description of the horses/experimental subjects and their clinical presentations.
I am not sure that this study really constitutes a validation of the equipment or its application, since data are incomplete and there are too many questions concerning the population studied. One of the issues may be of how heart sounds were evaluated for acoustic range and how easily they could be heard, and the need for establishment of equine “standards". I strongly recommend that no claim of validation be made (Line 458).
Although it becomes clear as the paper advances, it would be helpful to have a short description of what a "snapshot" might be early in the paper. The idea of continuous monitoring for several hours overnight is clear.
It would be helpful to have a table that showed the echocardiographic variables on one side and the Audicor® variables that were seen to be approximately equivalent, on the other, perhaps with brief information on why these are considered similar, if that is not obvious. This should appear in materials and methods, and might be more valuable in the context of the study than the tables of echocardiographic variables, which could become supplementary.
Conserning the issue of the "strength" of heart sounds, there are lots of variables involved including the species being examined, and for the horse an issue of great significance is likely to be the frequency spectrum of the sounds. At low frequencies sounds are as much "sensed" as palpable vibrations as they are heard, since they fall below the human auditory range. This may be an example of the dangers involved in applying equipment designed for another species and deserves greater attention.
Perhaps the discussion could have been written in the context of the relationship between heart sounds and hemodynamic events rather than as here, where hypotheses and interpretations of conventionally derived variables take priority. This accentuates the problem in the study in which so much of the interpretation depends on information to which the authors did not have access. Discussion of the relationships between heart sounds and hemodynamic events would have provided fundamental information that could help guide the development of equine algorithms and equine-oriented technologies.
Title. "Quantification of cardiac function in horses with atrial fibrillation before and after cardioversion using ambulatory acoustic cardiography." Does this fit? It serves the narrative of the present paper structure well, but a change in emphasis might require change. A suggestion would be “Application of acoustic cardiography in assessment of cardiac function in horses with atrial fibrillation before and after cardioversion”.
Line 68. This doesn't seem surprising since Audicor® directly measures this parameter, standard ultrasound does not, while measurement of ejection fraction depends on a number of assumptions. Measurements can be taken of electromechanical activation time, locally, with ultrasound. Perhaps the human benefits of the Audicor® measurement reflect the global nature of this approach (encompassing the entirety of ventricular activation and contraction up to S1)?
Line 86. This sentence is problematic. Science generally proceeds on the basis of the null hypothesis, that is, that there is no effect, and that if an effect its found all reasonable attempts are made to identify an effect cause other than that tested. This isn't always easy in clinical medicine. However, this sentence might be rewritten as something like "The objective was to evaluate its use in a clinical setting and its possible benefits as an adjunctive to echocardiography".
Line 92. Which raises the question of whether an S4 would be expected during AF, plus Line 206 says the Audicor® device seems to ignore S4 under some circumstances. Can you clarify?
Line 115. Does this mean that all horses were in NSR at the time of day-1 and day 2-7 follow-up?
Line 138. How often was this not possible and what sorts of problems precluded the process?
Line 142. Might this more accurately be described as instantaneous heart rate, especially if the same procedure was used for AF and converted horses?
Lines 165-178. Please state units and use parentheses in the equations to indicate mathematical hierarchy.
Line 203. This statement is potentially quite fundamental in the context of cross species applications of the technology and considering the range of body dimensions in the species stated, plus the possibility of wide differences in the amplitude of lower frequency components. Since this is a personal communication (a claim from the manufacturers) and not research performed by the authors and since the statement refers to filtered signals, perhaps it would be advisable to change the wording such that it is explicitly stated that this is untested. This might be particularly significant since S4 was not identified. The statement that the sound is not identified without association with a QRS is quite troubling.
Line 220. Please provide an explicit description of a "snapshot analysis". I assume this means an averaging of the best five cycles recorded, but I may be mistaken.
Linings 224-231. Difficult to follow considering the fact that human algorithms and thresholds appear to have been used to process and interpret equine data and no information can be provided (proprietary software).
Line 228 and Table S4. Citation used in Table S4 (and all supplementary tables) should be to citations in the main text.
"Snapshots" seems to be variously defined at different places in the paper. In this section the reference appears to be to a time window of 10 seconds, elsewhere it appears to be to individual cycles. Please clarify precisely which protocol was used. Please also describe the procedure used to select maximum versus median values and justify this. For S3 & S4, this is presumably the "Strength" variable described for the Audicor®. Since this valuable is included in the table, more information is required on this scale. For example, what does a score of 10 mean? How reproducible and meaningful is this score if snapshots are being selected on the basis of these heart sounds in the first instance?
What was the experimental design for the determination of between day variability?
Frequent reference is made to S3 and S4, but none to S1 and S2, yet these sounds must have been identified to generate several of the variables measured. What special considerations, if any, applied to the recognition of these sounds and how were the fiducial points identified?
Line 223. Systolic Dysfunction Index is included in the parameters measured and is a parameter derived as an overall indication of systolic function in humans. What is the extent to which this represents a valid parameter to derive from the equine data gathered?
Line 239. Reference is made variously to the strength and occasionally to power of S3 and S4 - are these meant to be the same and in what units are they measured?
Line 232-250. This paragraph might be better as a simple acknowledgement that this facility is included in the technology. It appears that application of this analysis was not always possible in the present study, while the paragraph provides insufficient information to objectively evaluate what is actually being achieved. Justifications for the thresholds employed (for example, SDI greater than or equal to 5 and 7.5) cannot be interpreted. The fact that these measurements have been approved by the FDA is encouraging, but since their derivation is unavailable and entirely based on human analysis, the specifics have limited value in the present context.
Line 271. "Summary statistics were provided…"
Line 281. Please consider whether "corresponding" or instead, perhaps "comparable" or "approximately equivalent" is most accurate.
Line 286. Thank you for including the group numbers in Table 1. The number of echocardiograms that were not "analysable" seems quite large considering the source of these data. I suggest a short sentence explaining why some were not suitable.
Table 1. The title of this table might be changed to something like "Summary statistics for heart rate and echocardiographic variables of left atrial size and function, by treatment stage."
Table 2. The title of this table might be changed to something like "Summary statistics for echocardiographic variables of left ventricular size and function, by treatment stage."
Line 418. Please change R2 to superscript 2 throughout to be consistent with other references to this statistic and usual nomenclature.
Line 458 and succeeding paragraphs. I'm not comfortable with the idea that this study validates the use of Audicor® to document and follow changes in chamber function after cardioversion. There is certainly the potential for this to be the case, but a more coherent study design with less missing information and targeted investigation of key variables would be required, alongside echocardiography, to justify such a statement. The study does pave the way for such an investigation.
Line 631. The list of abbreviations appears to be corrupted and requires attention. There are also errors in this list that require correction, for example, LVEPP
Figure S2. What filtering is applied to the biological signals? The heart sounds appear to be selected according to their maximum amplitude. What is this figure meant to illustrate? The axes are not labelled (X, Y, Z). How is the amplitude of the heart sounds standardized for measurement? Why do the measurements not encompass all of the sounds rather than just the peak?
Table S4. RC - since you have used a particular definition of this statistic, please provide a definition at first use and a reference.
Figure 2. The value of these plots might be increased further by including the mean bias on each graph (as a number) and the repeatability coefficient.
Author Response
Response to Reviewer 3 Comments
- Summary
Thank you very much for taking the time to review this manuscript. Please find the detailed responses and the corresponding revisions in track changes in the re-submitted files and in the point-by-point response. Below we address your main points and explain how we have revised the manuscript accordingly.
- Responses to Major Comments
- a) Echocardiographic Data Emphasis and Manuscript Complexity
Comment: “Conversely, there is a degree of emphasis on the echocardiographic measurements taken and their demonstration of chamber dysfunction in AF cases after cardioversion that exceeds that necessary to achieve the paper’s primary objective. As a result, the paper is long and far more complex than necessary.”
Response: We appreciate the reviewer´s concern regarding the level of detail provided on echocardiographic measurements and manuscript length and complexity. The comprehensive echocardiographic assessment serves the critical purpose of characterizing the study sample using a well-established modality and of providing a robust comparison with the novel variables derived from acoustic cardiography. This comparison is central to evaluating the potential of the Audicor® device as an adjunctive tool and supports interpretation of its clinical relevance in the context of equine atrial fibrillation.
However, we recognize the importance of maintaining manuscript clarity and conciseness and are amenable to relocating detailed echocardiographic tables (Table 2 and 3) to supplementary materials while retaining summary statistics in the main text. We respectfully leave this decision to the editor´s discretion.
Comment: “Ironically, the authors conclude that the device's performance fell short in this regard.”
Response: We acknowledge the reviewer’s observation about this apparent contradiction. However, we respectfully suggest that identifying performance limitations was always a possible and scientifically valuable outcome of this proof-of-concept study. Demonstrating where current technology falls short provides important insights into its current capabilities, realistic expectations for clinical implementation and guides future optimization. Rather than diminishing the study´s value, these findings contribute crucial insights into the current state of acoustic cardiography in horses.
- b) Missing Data and Data Quality
Comment: “The analysis performed of the data collected is extensive and very thorough. However, there is also a lot of missing raw data, which combined with the lack of detail on the experimental subjects and on the protocols followed in the involved institutions, makes it difficult to assess the quality and relevance of the information gathered. In some instances the reasons for the missing data are addressed, in others these explanations are missing.”
Response: We agree that missing data represents a study limitation. These gaps reflect the inherent challenges of conducting a multicenter study in a clinical setting, where optimization of data quality across centers can be difficult to achieve. We have made additional efforts to more clearly address the reasons for missing data throughout the revised manuscript and have included a supplementary table (Table S1) detailing the number of analyzed cycles for each echocardiographic measurement at each time point. Please see our point-by-point responses to the relevant line-specific comments below for clarification and updates made within the manuscript text.
We hope these revisions enhance clarity and support the reviewer’s assessment of the study’s scope and limitations.
- c) Study Sample Description
Comment: “The data used are multicenter convenience data. This is a valid approach, though little information is presented on patent specifics. Greater detail is required on the description of the horses/experimental subjects and their clinical presentations.”
Response: We recognize the importance of comprehensive population characterization for study reproducibility and clinical applicability. Our current description covers key clinical variables such as breed, age, sex, body weight, athletic use, cardiac findings, inclusion/exclusion criteria and core echocardiographic assessment including key variables describing LA and LV size and function.
This study investigated if Audicor® can detect and quantify atrial mechanical dysfunction, impaired ventricular function, as well as recovery thereof after successful conversion of AFib to NSR by comparing Audicor® recordings and concomitant echocardiographic recordings. However, the study was not aimed to identify horse-related factors influencing cardiac size and function after treatment of AFib. Therefore, we strongly believe that the information describing the study sample is sufficient for this particular study and that more data on individual horses are not necessary, would not strengthen this study and would neither change the outcome nor increase the applicability of results.
- d) Validation claims
Comment: “I am not sure that this study really constitutes a validation of the equipment or its application, since data are incomplete and there are too many questions concerning the population studied. One of the issues may be of how heart sounds were evaluated for acoustic range and how easily they could be heard, and the need for establishment of equine “standards". I strongly recommend that no claim of validation be made (Line 458).”
Response: We agree that the study does not constitute a formal validation of the Audicor® device. In line with this recommendation, we have revised the manuscript to remove all validation claims throughout the text (including former line 458). Instead, we now refer to this study as a comparative assessment of Audicor® parameters against conventional echocardiographic indices. This updated phrasing more accurately reflects the scope and limitations of our study and avoids overstating the findings while appropriately positioning this as preliminary technology evaluation in horses.
- e) “Snapshot” Description
Comment: “Although it becomes clear as the paper advances, it would be helpful to have a short description of what a "snapshot" might be early in the paper. The idea of continuous monitoring for several hours overnight is clear.”
Response: A description of “snapshot” recordings was already included in the manuscript. However, we have now improved and clarified this section to provide a more concise explanation. Please see the revised text in the Materials and Methods section (page 4-5, lines 235-248), as well as the point-by-point response below.
- f) Comparative Table of Audicor® Variables and Echocardiographic Variables
Comment: “It would be helpful to have a table that showed the echocardiographic variables on one side and the Audicor® variables that were seen to be approximately equivalent, on the other, perhaps with brief information on why these are considered similar, if that is not obvious. This should appear in materials and methods, and might be more valuable in the context of the study than the tables of echocardiographic variables, which could become supplementary.”
Response: We appreciate this insightful suggestion and have included a new table in the Materials and Methods section (Table 1) that presents a side-by-side comparison of echocardiographic variables and corresponding Audicor® variables, along with brief explanations of their similarities and the rationale for comparison. This table enhances understanding of how acoustic cardiography variables relate to established echocardiographic measurements.
Regarding the echocardiographic variable tables (Table 2 and 3), we agree this is a reasonable option to consider moving them to supplementary material to streamline the main text, although we believe that the echo data are crucial for the comprehensive description of the study sample. We respectfully leave the final decision on this matter to the editor, while remaining fully prepared to relocate them if preferred.
- g) Species-Specific Frequency Spectrum Considerations
Comment: “Concerning the issue of the "strength" of heart sounds, there are lots of variables involved including the species being examined, and for the horse an issue of great significance is likely to be the frequency spectrum of the sounds. At low frequencies sounds are as much "sensed" as palpable vibrations as they are heard, since they fall below the human auditory range. This may be an example of the dangers involved in applying equipment designed for another species and deserves greater attention.”
Response: We acknowledge this important technical point and recognize that the species-specific frequency spectrum of heart sounds in horses presents unique challenges and limits the direct application of equipment originally developed for humans. This species-specific challenge highlights a fundamental limitation in cross-species application of acoustic cardiography technology. We have expanded our discussion of this limitation and its implications for future equine-specific device development, emphasizing the need for technologies specifically designed for equine cardiovascular acoustics (page 20-21, lines 637-667).
- h) Discussion Focus on Hemodynamic Relationships
Comment: “Perhaps the discussion could have been written in the context of the relationship between heart sounds and hemodynamic events rather than as here, where hypotheses and interpretations of conventionally derived variables take priority. This accentuates the problem in the study in which so much of the interpretation depends on information to which the authors did not have access. Discussion of the relationships between heart sounds and hemodynamic events would have provided fundamental information that could help guide the development of equine algorithms and equine-oriented technologies.”
Response: We thank the reviewer for his input regarding the relationship between heart sounds and hemodynamic events. While our study was designed as a comparative assessment to assess the device´s utility as an adjunctive tool rather than a mechanistic investigation, we recognize that understanding acoustic-hemodynamic relationships is fundamental for advancing equine acoustic cardiography. Such mechanistic analysis would require simultaneous pressure-volume measurements and detailed phonocardiographic studies. Approaches that, while scientifically valuable, represent distinct research objectives requiring different methodological frameworks from our current comparative study design.
We have acknowledged this important research direction in our Discussion section (page 21, lines 657-667) and emphasized how future mechanistic studies investigating the relationships between heart sounds and hemodynamic events could inform development of equine-specific algorithms and improve the clinical utility of acoustic cardiography in horses. This represents a critical avenue for advancing the field beyond simple comparative assessments.
- Point-by-point response to Minor Comments and Suggestions for Authors
Comment 1: Title "Quantification of cardiac function in horses with atrial fibrillation before and after cardioversion using ambulatory acoustic cardiography." Does this fit? It serves the narrative of the present paper structure well, but a change in emphasis might require change. A suggestion would be “Application of acoustic cardiography in assessment of cardiac function in horses with atrial fibrillation before and after cardioversion”.
Response 1: We agree that the proposed title more accurately reflects the focus and narrative structure of the manuscript. Therefore, we have revised the title accordingly.
“Application of acoustic cardiography in assessment of cardiac function in horses with atrial fibrillation before and after cardioversion”.
Comment 2: Line 68 This doesn't seem surprising since Audicor® directly measures this parameter, standard ultrasound does not, while measurement of ejection fraction depends on a number of assumptions. Measurements can be taken of electromechanical activation time, locally, with ultrasound. Perhaps the human benefits of the Audicor® measurement reflect the global nature of this approach (encompassing the entirety of ventricular activation and contraction up to S1)?
Response 2: We agree that the ability of Audicor® to capture global electromechanical activation, as opposed to more assumption-based regional estimates of function by echocardiography, may explain its sensitivity in detecting early dysfunction. As this sentence appears in the introduction and refers to findings in human studies [33], we have left the text unchanged, but we appreciate the opportunity to reflect on the physiological principles that may underlie these differences.
Comment 3: Line 86 This sentence is problematic. Science generally proceeds on the basis of the null hypothesis, that is, that there is no effect, and that if an effect its found all reasonable attempts are made to identify an effect cause other than that tested. This isn't always easy in clinical medicine. However, this sentence might be rewritten as something like "The objective was to evaluate its use in a clinical setting and its possible benefits as an adjunctive to echocardiography".
Response 3: We agree with the reviewer´s concern regarding the phrasing and have revised the sentence to better reflect the exploratory nature of the study (page 1, line 88-89).
“The objective was to evaluate its use in a clinical setting and its possible benefits as an adjunctive to echocardiography.”
Comment 4: Line 92 Which raises the question of whether an S4 would be expected during AF, plus Line 206 says the Audicor® device seems to ignore S4 under some circumstances. Can you clarify?
Response 4:
We agree that, obviously, S4 is not expected during atrial fibrillation, as its generation relies on a coordinated atrial contraction, which is absent in AF. This understanding underlies our hypothesis. As this is a well-established physiological concept, we have not added further explanation to the manuscript, but we appreciate the opportunity to clarify our reasoning here. We have now also clarified in the manuscript that S4 powers below 5 are primarily considered artefactual (see page 5, line 249-253). Therefore, we have not altered the original hypothesis in the introduction but hope this provides the necessary clarification.
Comment 5: Line 115 Does this mean that all horses were in NSR at the time of day 1 and day 2-7 follow-up?
Response 5: We confirm that all horses were in NSR at day 1 of follow-up. For the 2-7 day follow-up period, only horses that remained in NSR throughout this interval were included in the analysis. If a horse reverted to AF before this time point, its data were excluded. This clarification has now been incorporated into the manuscript (page 2, line 118-120).
“For the 2-7 follow-up period, only data from horses that remained in NSR throughout this time period were included in the analysis, whereas data from horses that reverted to AF before this time point were excluded.”
Comment 6: Line 138 How often was this not possible and what sorts of problems precluded the process?
Response 6: We have now clarified this aspect in the Material and Methods section and have included a supplementary table (Table S1), which summarizes the number of echocardiographic measurements that were based on fewer than three cardiac cycles. The primary reasons for obtaining fewer than three analyzable cycles were related to technical limitations during image acquisition, such as brief data recordings, suboptimal echocardiographic image quality, or in case of color tissue doppler imaging (cTDI) suboptimal alignment of the doppler angle alignment relative to myocardial motion. These factors limited the number of cycles suitable for analysis. This clarification has now been incorporated into the manuscript (page 3, line 149-153).
“Where possible, three representative non-consecutive or consecutive cycles were recorded, measured and subsequently averaged for each variable. However, in some instances, fewer than three analyzable cardiac cycles were available due to technical limitations during image acquisition, such as suboptimal image quality. The number of measurements based on fewer than three cardiac cycles is summarized in Supplementary Table S1.”
Comment 7: Line 142 Might this more accurately be described as instantaneous heart rate, especially if the same procedure was used for AF and converted horses?
Response 7: We agree with the reviewer´s suggestion and have revised the text accordingly (page 3, line 155).
“Instantaneous HR was calculated based on the RR interval (ms) preceding the respective measurement: HR = 60000 / RR.”
Comment 8: Line 165-178 Please state units and use parentheses in the equations to indicate mathematical hierarchy.
Response 8: We have now added the appropriate units to clarify the reported values (page 3, line 178, 180, 182). Regarding the use of parentheses, we respectfully note that according to standard mathematical conventions (order of operations), exponents are evaluated prior to multiplication and division, and therefore the current notation is mathematically unambiguous and correct. Therefore, no changes have been applied.
“Chamber diameter [500] (cm) = Measured chamber diameter / BWT1/3 x 5001/3
Chamber area [500] (cm2) = Measured chamber area / BWT2/3 x 5002/3
Chamber volume [500] (mL) = Measured chamber volume / BWT x 500.”
Comment 9: Line 203 “This statement is potentially quite fundamental in the context of cross species applications of the technology and considering the range of body dimensions in the species stated, plus the possibility of wide differences in the amplitude of lower frequency components. Since this is a personal communication (a claim from the manufacturers) and not research performed by the authors and since the statement refers to filtered signals, perhaps it would be advisable to change the wording such that it is explicitly stated that this is untested. This might be particularly significant since S4 was not identified. The statement that the sound is not identified without association with a QRS is quite troubling.
Response 9: We fully agree that cross-species differences, especially in regard to signal filtering, body size, and frequency spectrum, may affect the applicability of the technology and that further research is warranted in this area. In the revised manuscript, we have now explicitly clarified that these statements are based on personal communication with the manufacturer and that no published experimental data is currently available for interspecies comparisons between humans and horses. We have also emphasized that the absence of S4 detection when unassociated with a QRS complex represents a known technical limitation of the current algorithm, which warrants further investigation (Material and Methods, page 4, line 216-225). Furthermore, we have expanded on this point in the Discussion section, under limitations (page 20, line 637-647), to emphasize that the absence of experimental validation presents a constraint on the generalizability of the findings and underlines the need for further species-specific algorithm development.
“According to information provided by the manufacturer (Inovise Medical Inc., personal communication), comparison of the filtered signals from dogs, pigs, horses, and other animals to those of humans does not show any fundamental differences concerning their frequency content and the placement of the relevant fiducial points However, no experimental data are currently available to validate these cross-species comparisons in horses specifically. Importantly, S4 is not detected or quantified when it occurs in isolation and lacks association with a QRS-T complex (e.g., with a second-degree AV block or in context of atrial arrhythmia with AV blocks, representing a limitation of the current algorithm.”
“The main limitation of this study is that the Audicor® device and its proprietary signal processing algorithms were originally developed and validated for use in humans. Detection thresholds such as the cut-off of 5 for S3 and S4 power were derived from human validation studies and may not directly translate to horses. In particular, low-frequency components of equine heart sounds may require dedicated research to establish appropriate detection thresholds. Similarly, SDI, which was developed for use in human patients, was included in this study to allow initial exploration of its behavior in horses. However, its diagnostic value in the equine setting remains uncertain and warrants further investigation. The lack of equine-specific algorithm development limits the generalizability of the findings and may have affected the accuracy and sensitivity of heart sound detection and systolic time interval measurements.”
Comment 10: Line 220 Please provide an explicit description of a "snapshot analysis". I assume this means an averaging of the best five cycles recorded, but I may be mistaken.
Response 10: We have now expanded the description of the snapshot analysis in the Materials and Methods section (page 4, lines 235-241). In brief, a "snapshot analysis" refers to the evaluation of all analyzable cardiac cycles within a 10-second segment of the overnight recording, with individual cardiac cycles automatically detected and processed by the proprietary software. For each horse, five high-quality 10-second snapshots were selected (preferably between 8:00 pm and 9:00 pm), and relevant variables were extracted from these segments. The median or maximum values from these five snapshots were then used for subsequent analyses, as detailed in the revised text.
“For each overnight recording, five consecutive, good quality, analyzable 10- second snapshot analyses were generated. A snapshot refers to continuous 10-second segment of the recording during which all heartbeats are automatically detected and processed by the proprietary software. Snapshots were selected based on ECG signal quality within the time frame of 8:00 pm and 9:00 pm. If insufficient diagnostic snapshots were available in this time frame due to suboptimal recording quality, the next best five snapshots from the recording were selected.”
Comment 11: Line 224-231 Difficult to follow considering the fact that human algorithms and thresholds appear to have been used to process and interpret equine data and no information can be provided (proprietary software).
Response 11: We fully acknowledge that the S3 and S4 detection algorithms and thresholds are based on proprietary algorithms originally developed for human application. This represents a limitation when applying the system to equine patients, as species-specific algorithms and validation data are not yet available. As this was a proof-of-concept study, we applied the existing algorithms without species-specific adaptation to explore the general feasibility and clinical applicability of acoustic cardiography in horses. We have now clarified in the revised manuscript that S3 and S4 powers below 5 are considered artefactual according to the manufacturer’s software specifications, but that the appropriateness of these thresholds for horses remains unvalidated and should be interpreted with caution. This limitation has been more explicitly addressed in both the Methods (page 5, line 253-255) and Discussion (page 20, line 637-647) sections of the revised manuscript.
“Importantly, values below 5 for S3 and S4 are considered artefacts and do not correspond to actual heart sounds; this threshold was set by the software to ensure optimal accuracy compared to visual overread by phonocardiography experts and to define the actual presence of heart sounds in people (Inovise Medical Inc., personal communication) [35]. It should be noted that these detection algorithms and thresholds were originally developed and validated for human cardiac data and were applied here without species-specific adaptation.
“The main limitation of this study is that the Audicor® device and its proprietary signal processing algorithms were originally developed and validated for use in humans. Detection thresholds such as the cut-off of 5 for S3 and S4 power were derived from human validation studies and may not directly translate to horses. In particular, low-frequency components of equine heart sounds may require dedicated research to establish appropriate detection thresholds. Similarly, SDI, which was developed for use in human patients, was included in this study to allow initial exploration of its behavior in horses. However, its diagnostic value in the equine setting remains uncertain and warrants further investigation. The lack of equine-specific algorithm development limits the generalizability of the findings and may have affected the accuracy and sensitivity of heart sound detection and systolic time interval measurements.”
Comment 12: Table S4 Citation used in Table S4 (and all supplementary tables) should be to citations in the main text.
Response 12: We have now revised the citations in Table S4 and all supplementary tables to reference only citations already included in the main text, as recommended.
Comment 13: "Snapshots" seems to be variously defined at different places in the paper. In this section the reference appears to be to a time window of 10 seconds, elsewhere it appears to be to individual cycles. Please clarify precisely which protocol was used. Please also describe the procedure used to select maximum versus median values and justify this. For S3 & S4, this is presumably the "Strength" variable described for the Audicor®. Since this valuable is included in the table, more information is required on this scale. For example, what does a score of 10 mean? How reproducible and meaningful is this score if snapshots are being selected on the basis of these heart sounds in the first instance?”
Response 13: We have now clarified throughout the manuscript that a "snapshot" refers specifically to a 10-second continuous recording segment, during which all cardiac cycles are automatically identified and analyzed by the proprietary software. All reported snapshot analyses were based on this consistent 10-second window approach. Snapshots were selected exclusively based on ECG signal quality and not on the presence of S3 or S4, to avoid bias.
For each overnight recording, five analyzable snapshots were selected with a predefined time window. To improve reproducibility, the median value of EMAT, EMATc, LVST, LVSTc, and SDI across these five snapshots was used for further analysis. This approach was based on a reproducibility assessment (see Supplementary Table S5), which showed better reproducibility using the median of multiple snapshots compared to single-snapshot measurements. For S3 and S4, given the intermittent nature of these heart sounds, we selected the maximum power value across the five snapshots.
Regarding the power variable for S3 and S4, this is a proprietary dimensionless scale ranging from 1 to 10, which reflects both the intensity and persistence of the respective heart sound within a snapshot/overnight recording, as determined by the manufacturer's algorithm. The value represents a relative measure, rather than an absolute amplitude or physical unit, and was developed based on proprietary human datasets. A threshold of 5 was established by the manufacturer to differentiate true heart sounds from artifacts based on validation studies in humans. As the same algorithms were applied in this proof-of-concept study for horses, we acknowledge that species-specific validation of this scale has not yet been performed.
These clarifications have been incorporated into the revised Materials and Methods section (pages 4, lines 235-263), and we hope they address the reviewer’s concerns.
“For each overnight recording, five consecutive, good quality, analyzable 10-second snapshot analyses were generated. A snapshot refers to continuous 10-second segment, of the recording during which all heartbeats are automatically detected and processed by the proprietary software. Snapshots were selected based on ECG signal quality within the time frame of 8:00 pm and 9:00 pm. If insufficient diagnostic snapshots were available in this time frame due to suboptimal recording quality, the next best five snapshots from the recording were selected. The following variables were generated for each snapshot analysis: HR, EMAT, heart rate-corrected EMAT (EMATc), left ventricular systolic time (LVST), heart rate-corrected LVST (LVSTc), power (as a function of intensity and persistence) of S3, power h of S4, and systolic dysfunction index (SDI, as a function of QRS duration, QT interval, EMATc, and S3) [40] (Table S4). Power is calculated by the software on a relative dimensionless scale from 1 to 0, reflecting both signal intensity and persistence. Importantly, values below 5 for S3 and S4 are considered artefactual and do not correspond to true heart sounds; this threshold was set by the software to ensure optimal accuracy compared to visual overread by phonocardiography experts and to define the actual presence of heart sounds in people (Inovise Medical Inc., personal communication) [35]. It should be noted that these detection algorithms and thresholds were originally developed and validated for human cardiac data and were applied here without species-specific adaptation. To enhance measurement reproducibility, for EMAT, EMATc, LVST, LVSTc and SDI, the median value across the five selected snapshots was used for further analyses. This approach was chosen based on pilot data from a repeatability study (Table S5), which demonstrated improved reproducibility compared to relying on a single snapshot. For the power of S3 and S4, were presence may vary across snapshots, the maximum value across five consecutive snapshots was selected for further analyses”.
Comment 14: What was the experimental design for the determination of between day variability?
Response 14: We have clarified the experimental design in the revised supplementary materials (Table S5). Briefly, between-day variability was assessed in 10 randomly selected horses, each undergoing three repeated measurements on separate days. Variability was quantified using one-way ANOVA to determine within-subject variance, from which the within-subject standard deviation (sw), coefficient of variation (CV), and repeatability coefficient (RC) were calculated following established statistical methods. A detailed description has been added to the Supplementary S5 section.
“Test reproducibility was quantified based on the three repeated measurements obtained in each of the 10 randomly selected horses. Within-subject variance for repeated measurements (residual mean square), determined by a one-way ANOVA with horses as groups, was used for the quantification of reproducibility. The within-subject standard deviation (sw) was calculated as the square root of the residual mean square. Measurement variability was reported as coefficient of variation (CV) and as repeatability coefficient (RC) according to the British Standard Institution (BSI). The CV as a per cent value was calculated as CV = sw/mean x 100. The RC is the absolute value below which the difference between two measurements will lie with 95 % probability and is calculated as 1.96 x √2 x sw = 2.77 x sw [46]. The RC is clinically more applicable and standardized compared to the CV and it considers uncertainty of the point estimates and uncertainty of prediction associated with repeated measurements [47]. By comparing the RC to the magnitude of change observed in a variable over time, one can assess if the change is a result of measurement error and physiologic variability (observed change ≤RC) or a true change over time (observed change >RC) [40].”
Comment 15: Frequent reference is made to S3 and S4, but none to S1 and S2, yet these sounds must have been identified to generate several of the variables measured. What special considerations, if any, applied to the recognition of these sounds and how were the fiducial points identified?
Response 15: As correctly noted, identification of S1 and S2 is essential for generating timing-related parameters such as EMAT and LVST. However, in this study, S1 and S2 were not independently quantified for amplitude or strength. Their primary role was to serve as fiducial markers for systolic timing measurements.
The placement of fiducial points for S1 and S2 was performed automatically by the proprietary Audicor® software using combined analysis of the acoustic and ECG signals. The algorithms applied were originally developed and validated in human datasets and were used here without species-specific modifications. The lack of equine-specific algorithm adaptation represents a general limitation of the study, which has now been explicitly addressed in the revised discussion section (page 20, lines 645-647).
Comment 16: Line 223 Systolic Dysfunction Index is included in the parameters measured and is a parameter derived as an overall indication of systolic function in humans. What is the extent to which this represents a valid parameter to derive from the equine data gathered?
Response 16: The Systolic Dysfunction Index (SDI) was developed and validated for use in human patients, where it has demonstrated utility in assessing systolic function. However, its application in equine patients remains questionable, as a previous study has shown high day-to-day variability of SDI values in horses [40]. In the present proof-of-concept study, SDI was included to allow initial exploration of its behavior in the equine setting, while acknowledging that its diagnostic value in horses requires further investigation (page 20, line 642-645).
“Similarly, SDI, which was developed for use in human patients, was included in this study to allow initial exploration of its behavior in horses. However, its diagnostic value in the equine setting remains uncertain and warrants further investigation. “
Comment 17: Line 239 Reference is made variously to the strength and occasionally to power of S3 and S4 - are these meant to be the same and in what units are they measured?
Response 17: The variable previously referred to as “Strength” has now been uniformly termed “Power” throughout the manuscript for consistency and clarity. This variable represents a dimensionless proprietary index reflecting the combination of sound intensity and persistence as determined by the Audicor® system.
Comment 18: Line 232-250 This paragraph might be better as a simple acknowledgement that this facility is included in the technology. It appears that application of this analysis was not always possible in the present study, while the paragraph provides insufficient information to objectively evaluate what is actually being achieved. Justifications for the thresholds employed (for example, SDI greater than or equal to 5 and 7.5) cannot be interpreted. The fact that these measurements have been approved by the FDA is encouraging, but since their derivation is unavailable and entirely based on human analysis, the specifics have limited value in the present context.
Response 18: We fully acknowledge that the “cardiac findings” report, including the specific thresholds and cut-off values, was developed and validated for use in human patients and is based on proprietary algorithms approved by the FDA. As this was a proof-of-concept study, we applied the existing analysis framework without species-specific adaptation to explore the general feasibility of acoustic cardiography in horses before and after cardioversion. This limitation has now been clearly acknowledged in the revised Discussion section (page 20, lines 637-647).
Comment 19: Line 271 "Summary statistics were provided…"
Response 19: The text has been corrected to use the correct verb form.
“Summary statistics were provided using mean and standard deviation (SD).”
Comment 20: Line 281 Please consider whether "corresponding" or instead, perhaps "comparable" or "approximately equivalent" is most accurate.
Response 20: We have replaced “corresponding” with “comparable”.
“Agreement between comparable TDI variables and Audicor® variables was evaluated using Bland-Altman analyses.”
Comment 21: Line 286 Thank you for including the group numbers in Table 1. The number of echocardiograms that were not "analysable" seems quite large considering the source of these data. I suggest a short sentence explaining why some were not suitable.
Response 21: We agree that the number of non-analyzable echocardiograms was relatively high. This was primarily due to the multicenter study design, where standardization and optimization of image acquisition across centers posed certain challenges and led to variability in image quality. We have now added a sentence to the manuscript to clarify this (page 7, line 333-335).
“Non-analyzable echocardiograms were primarily attributable to the multicenter study design, which made consistent optimization of image quality across centers challenging.”
Comment 22: Table 1. The title of this table might be changed to something like "Summary statistics for heart rate and echocardiographic variables of left atrial size and function, by treatment stage."
Response 22: We agree with the reviewer’s suggestion and have adjusted the table title accordingly. As the grouping was based on predefined time points rather than treatment stage, the title was changed to: “Summary statistics for heart rate and echocardiographic variables of left atrial size and function, by time point.”
Comment 23: Table 2. The title of this table might be changed to something like "Summary statistics for echocardiographic variables of left ventricular size and function, by treatment stage."
Response 23: We agree with the reviewer’s suggestion and have adjusted the table title accordingly. As the grouping was based on predefined time points rather than treatment stage, the title was changed to: “Summary statistics for echocardiographic variables of left ventricular size and function, by time point.”
Comment 24 Line 418 Please change R2 to superscript 2 throughout to be consistent with other references to this statistic and usual nomenclature.
Responds 24: All instances of R2 have been revised to use superscript 2 (R²) throughout the manuscript to ensure consistency with standard scientific nomenclature.
Comment 25: Line 458I'm not comfortable with the idea that this study validates the use of Audicor® to document and follow changes in chamber function after cardioversion. There is certainly the potential for this to be the case, but a more coherent study design with less missing information and targeted investigation of key variables would be required, alongside echocardiography, to justify such a statement. The study does pave the way for such an investigation.
Response 25: We agree that the study does not constitute a formal validation of the Audicor® device. In line with this recommendation, we have revised the manuscript to remove all validation claims throughout the text (including former line 458). Instead, we now refer to this study as a comparative assessment of Audicor® parameters against conventional echocardiographic indices. This updated phrasing more accurately reflects the scope and limitations of our study and avoids overstating the findings while appropriately positioning this as preliminary technology evaluation in horses.
Comment 26: Line 631The list of abbreviations appears to be corrupted and requires attention. There are also errors in this list that require correction, for example, LVEPP
Response 26: The list has been thoroughly reviewed and corrected to fix formatting issues and errors.
Comment 27: Figure S2. What filtering is applied to the biological signals? The heart sounds appear to be selected according to their maximum amplitude. What is this figure meant to illustrate? The axes are not labelled (X, Y, Z). How is the amplitude of the heart sounds standardized for measurement? Why do the measurements not encompass all of the sounds rather than just the peak?
Response 27: The purpose of Figure S3 is primarily illustrative, to demonstrate the relationship between the simultaneously recorded electrocardiogram and phonocardiogram signals and to visualize the automatic placement of fiducial points used for systolic time interval calculations (e.g. EMAT and LVST) by the proprietary Audicor® algorithm. As indicated in the revised figure legend, the fiducial points for S1, S2, and S4 are automatically placed at the peaks of the respective sound amplitudes by the algorithm. No independent amplitude standardization was applied, as the signal processing and filtering are performed through the proprietary algorithm originally developed for human data. The axes are labelled directly in the figure: time on the horizontal axis, acoustic energy on the y-axis, and frequency on the z-axis for the 3D representation. We have now clarified these points and expanded the figure legend accordingly to provide better transparency on signal processing and the intended illustrative purpose of the figure.
“The fiducial points for S1, S2 and S4 are placed at the respective peak amplitudes of the heart sounds, as identified by the algorithm. Amplitude standardization and signal filtering are based on proprietary processing algorithms developed for human data. The figure serves as an illustration of signal alignment and the derivation of systolic time intervals, rather than for quantitative evaluation.”
Comment 28: Table S4. RC - since you have used a particular definition of this statistic, please provide a definition at first use and a reference.
Response 28: We have now included a detailed definition and explanation of the repeatability coefficient (RC) at its first use in Supplement S5.
“Measurement variability was reported as coefficient of variation (CV) and as repeatability coefficient (RC) according to the British Standard Institution (BSI). The CV as a per cent value was calculated as CV = sw/mean x 100. The RC is the absolute value below which the difference between two measurements will lie with 95 % probability and is calculated as 1.96 x √2 x sw = 2.77 x sw [46]. The RC is clinically more applicable and standardized compared to the CV and it considers uncertainty of the point estimates and uncertainty of prediction associated with repeated measurements [47]. By comparing the RC to the magnitude of change observed in a variable over time, one can assess if the change is a result of measurement error and physiologic variability (observed change ≤RC) or a true change over time (observed change >RC) [40].”
Comment 29: Figure 2. The value of these plots might be increased further by including the mean bias on each graph (as a number) and the repeatability coefficient.
Response 29: The mean bias is already indicated in each graph as the solid line and corresponding numeric value. The repeatability coefficient is available for the Audicor® measurements and is presented in the Supplement (Table S5). Since the repeatability coefficient was not available for the echocardiographic reference measurements, we have opted not to include it directly in Figure 2 to ensure consistency across all plots.
Round 2
Reviewer 3 Report
Comments and Suggestions for Authors
I am happy that the authors have responded appropriately and fully to the recommendations made and that the paper should go forward for publication. I remain concerned that the discussion of electrical and mechanical dysfunction in equine atrial fibrillation is more extensive than necessary and could be dealt with in much less space, though this would not change the specific references to the literature and is perhaps a matter of style. My thanks to the authors for their thoroughness.